# Complementary Label Learning with Positive Label Guessing and Negative Label Enhancement

**Yuhang Li**[1]  **Zhuying Li**[1]  **Yuheng Jia**[1,2] *

[1]School of Computer Science and Engineering, Southeast University, Nanjing 210096, China
[2]Key Laboratory of New Generation Artificial Intelligence Technology and Its Interdisciplinary
Applications (Southeast University), Ministry of Education, China
`{yuhangli, zhuyingli, yhjia}@seu.edu.cn`

## Abstract

Complementary label learning (CLL) is a weakly supervised learning paradigm that constructs a multi-class classifier only with complementary labels, specifying classes that the instance does not belong to. We reformulate CLL as an inverse problem that infers the full label information from the output space information. To be specific, we propose to split the inverse problem into two subtasks: *positive label guessing* (PLG) and *negative label enhancement* (NLE), collectively called PLNL. Specifically, we use well-designed criteria for evaluating the confidence of the model output, accordingly divide the training instances into three categories: highly-confident, moderately-confident and under-confident. For highly-confident instances, we perform PLG to assign them pseudo labels for supervised training. For moderately-confident and under-confident instances, we perform NLE by enhancing their complementary label set at different levels and train them with the augmented complementary labels iteratively. In addition, we unify PLG and NLE into a consistent framework, in which we can view all the pseudo-labeling-based methods from the perspective of negative label recovery. We prove that the error rates of both PLG and NLE are upper bounded, and based on that we can construct a classifier consistent with that learned by clean full labels. Extensive experiments demonstrate the superiority of PLNL over the state-of-the-art CLL methods, e.g., on STL-10, we increase the classification accuracy from 34.96% to 55.25%. The source code is available at https://github.com/yhli-ml/PLNL.

## 1 Introduction

Over the past few years, large-scale and accurately labeled data has tremendously boosted the development of deep neural networks. However, collecting accurately labeled data is extremely time-consuming, labor-intensive and sometimes requires specific expertise in real-world tasks. To reduce the dependency on large-scale and accurately labeled datasets, deep learning communities have given increasing attention to weakly supervised learning, including but not limited to partial label learning (Cour et al., 2011; Xie and Huang, 2018; Feng and An, 2019; Lv et al., 2020; Xia et al., 2023; Jia et al., 2023; 2024; Yang et al., 2024; Huang and Cheung, 2024; He et al., 2024; Tian et al., 2024), noisy label learning (Natarajan et al., 2013; Han et al., 2018; Song et al., 2023; Wei et al., 2020; Zhang et al., 2023; Huang et al., 2023), semi-supervised learning (Van Engelen and Hoos, 2020; Sohn et al., 2020; Xie et al., 2020; Yang et al., 2023; Li et al., 2024; Jiang et al., 2024b), positive-unlabeled learning (Niu et al., 2016; Kiryo et al., 2017).

Here, we consider a recently proposed weakly supervised learning framework called complementary label learning (CLL) (Ishida et al., 2017; Feng et al., 2020). In CLL, each training instance is associated with one or multiple complementary labels (CLs) which specify one or multiple classes that the instance does not belong to. The goal of CLL is to learn a multi-class classifier only from complementary labeled data. In real-world scenario, if the number of classes is huge, choosing the

---
*Corresponding author.

correct class label from many candidate classes is difficult and laborious, while choosing one or several of the incorrect class labels as CLs would be much easier and thus less costly. Recently, CLL has been applied to online learning (Kaneko et al., 2019), medical image segmentation (Rezaei et al., 2020) and medical molecular imaging (Tapper et al., 2024), etc. Besides, another promising future application scenario of CLL is to ensure privacy security in data collection scenarios. For example, collecting some survey data may require extremely private questions and it would be mentally less demanding if we ask the respondent to provide some incorrect answers as CLs (Dwork, 2008).

Previous studies on CLL can be roughly divided into two categories: methods that attempt to construct an unbiased risk estimator (URE-based) (Ishida et al., 2017; 2019; Feng et al., 2020) and methods based on feature learning (FL-based) (Chou et al., 2020; Wang et al., 2021; Liu et al., 2022; Jiang et al., 2024a). For URE-based methods, Ishida *et al.* (Ishida et al., 2017) and Feng *et al.* (Feng et al., 2020) showed that the ordinary classification risk can be recovered by their proposed unbiased risk estimator only from complementary labeled data. Ishida *et al.* (Ishida et al., 2019) later extended the unbiased risk estimator to arbitrary losses and models. Chou *et al.* (Chou et al., 2020) proposed a surrogate complementary loss framework, which avoids the extremely noisy gradient problem encountered in unbiased risk estimator. For FL-based methods, Wang *et al.* (Wang et al., 2021) gave the first attempt to leverage regularization techniques with complementary label by aligning the model output of one instance and its multiple augmented views. Liu *et al.* (Liu et al., 2022) proposed to integrate self-supervised and self-distillation to complementary learning. Jiang *et al.* (Jiang et al., 2024a) leveraged a contrastive learning framework to facilitate CLL. These methods mainly focus on the design of robust loss functions or the exploration of feature space information, while neglecting the power of output space information.

We propose that CLL can be viewed as solving the multi-class classification problem from two inverse aspects, where one is to infer the positive label and another is to infer the negative labels. To this end, we propose two subtasks: *positive label guessing* (PLG) and *negative label enhancement* (NLE). We use well-designed criteria for evaluating the confidence of the model output, accordingly divide the training instances into three categories: highly-confident, moderately-confident and under-confident in each epoch. We perform PLG by simply pseudo-labeling highly-confident instances for supervised training. Unlike pseudo-labeling methods used in semi-supervised learning (SSL), PLG pseudo-labeling reaches high selected ratio and high precision even without any positive labels available.

More importantly, previous SSL methods lack the utilization of untrustworthy instances. They either discard this part or simply employ techniques such as consistency regularization. In this paper, we perform NLE by enhancing the negative label set of moderately-confident and under-confident instances and train them with the augmented negative labels iteratively.

Although PLG and NLE will inevitably bring pseudo-labeling errors, we theoretically prove that the error rates are upper bounded. And the generalization error of the learned classifier under PLG and NLE errors is also upper bounded, which means that we can construct a classifier consistent with that learned by clean full labels. We demonstrate that PLNL achieves state-of-the-art performance on five benchmark datasets. Our contributions can be summarized as follows:

- *A novel method for CLL.* Different from conventional loss design methods, we pioneer a novel method for CLL called PLNL that formulates CLL from output space information and solve it by two subtasks: PLG and NLE.

- *A unified framework for pseudo-labeling-based methods.* From the perspective of negative label recovery, we construct a unified framework for pseudo-labeling-based methods. We empirically show that PLNL outperforms state-of-the-art SSL methods in terms of pseudo-labeling error, selected ratio and recovered negative labels.

- *Solid theoretical analysis.* We theoretically prove that both the error rates of PLG and NLE are upper bounded. The generalization error of the learned classifier is also upper bounded.

- *State-of-the-art performance.* Extensive experiments on five benchmark datasets demonstrate the superiority of PLNL over the state-of-the-art CLL methods.

## 2 PRELIMINARIES

**Ordinary Multi-Class Classification.** Let $\mathcal{X} \in \mathbb{R}^d$ denote the feature space with $d$ dimensions and $\mathcal{Y} = \{1, 2, ..., K\}$ denote the label space with $K$ classes. The precisely labeled instance $(\boldsymbol{x}, y) \in \mathcal{X} \times \mathcal{Y}$ is sampled from an unknown joint probability distribution $P(\boldsymbol{X}, \boldsymbol{Y})$. The goal of ordinary multi-class classification is to learn a parameterized function $f(x) : \mathbb{R}^d \to \mathbb{R}^K$ that minimizes the classification risk:

$$R(f) = \mathbb{E}_{P(\boldsymbol{X}, \boldsymbol{Y})} \mathcal{L}(f(\boldsymbol{x}), y), \tag{1}$$

where $\mathbb{E}_{P(\boldsymbol{X}, \boldsymbol{Y})}$ refers to the expectation across all possible samples drawn from the distribution $P(\boldsymbol{X}, \boldsymbol{Y})$, $\mathcal{L} : \mathbb{R}^K \times \mathcal{Y} \to \mathbb{R}$ is a multi-class classification loss function. In this paper, we consider a common case where the function $f$ is a deep neural network with the softmax output layer, where $f(\boldsymbol{x})$ is considered as the output prediction confidence of the model on each class. Since the probability distribution $P(\boldsymbol{X}, \boldsymbol{Y})$ is unknown, we use the empirical risk $\widehat{R}(f)$ to approximate $R(f)$. Assuming a dataset $\{(\boldsymbol{x}_i, y_i)\}_{i=1}^N$ is independently drawn from distribution $P(\boldsymbol{X}, \boldsymbol{Y})$, then we have

$$\widehat{R}(f) = \frac{1}{N} \sum_{i=1}^N \mathcal{L}(f(\boldsymbol{x}), y_i). \tag{2}$$

**Complementary Label Learning.** Different from the ordinary multi-class classification, in CLL, let $\{(\boldsymbol{x}_i, \bar{Y}_i)\}_{i=1}^N$ be the complementary labeled dataset, where $N$ is the dataset size, $\bar{Y}_i$ indicates the complementary (negative) label set of $\boldsymbol{x}_i$. Each complementary labeled instance $(\boldsymbol{x}, \bar{Y}) \in \mathcal{X} \times \mathcal{Y}$ is sampled from an unknown joint probability distribution $\bar{P}(\boldsymbol{X}, \bar{\boldsymbol{Y}})$. Our goal is to learn a classifier that minimizes the classification risk Eq. (1) only from complementary labeled training instances. Then the empirical risk becomes:

$$\widehat{R}(f) = \frac{1}{N} \sum_{i=1}^N \bar{\mathcal{L}}_{CLL}(f(\boldsymbol{x}_i), \bar{Y}_i), \tag{3}$$

where $\bar{\mathcal{L}}_{CLL}$ is a specially designed loss function for learning from only complementary labeled data.

## 3 PROPOSED METHOD

The overall framework of PLNL is shown in Fig.1, and the pseudo-code is presented in Appendix A. We begin by employing weak and strong augmentation to a complementary labeled image $\boldsymbol{x}_i$, which leads to two augmented views $\boldsymbol{x}_i^w, \boldsymbol{x}_i^s$. These two images are then fed into a two-view network with shared weight $f(\boldsymbol{x}; \Theta)$ to obtain two prediction confidences $f(\boldsymbol{x}_i^w)$ and $f(\boldsymbol{x}_i^s)$. Then we utilize the two-view prediction confidences to select three subsets of training instances mentioned above, i.e., highly-confident, moderately-confident and under-confident. We select these subsets using the historical confidences of the previous training epochs to better alleviate confirmation bias. Finally, different techniques are utilized to conquer individual subsets. In this section, we first explain the well-designed confidence-based instances selection strategy, and then introduce the PLG for the highly-confident instances and two different versions of NLE for the moderately-confident instances and the under-confident instances in detail.

### 3.1 CONFIDENCE-BASED INSTANCES SELECTION

We first maintain two memory banks $\boldsymbol{M}^w$ and $\boldsymbol{M}^s$ for weak and strong augmentation respectively, each with size $t \times N \times K$ to store the historical prediction confidence over the past $t$ epochs. For simplification, we only consider one view here unless otherwise specified.

$$\boldsymbol{M}_i = [f^1(\boldsymbol{x}_i), \ldots, f^t(\boldsymbol{x}_i)], \tag{4}$$

where $f^t(\boldsymbol{x}_i)$ denotes the prediction confidence of the $t$-th epoch in the memory bank $\boldsymbol{M}$.

We propose to select subsets based on the following criteria.

$$\omega_{i1} = \forall 1 \leq j \leq t, \arg\max(f^j(\boldsymbol{x}_i)) \notin \bar{Y}_i, \tag{5}$$

$$\omega_{i2} = \forall 1 \leq j, k \leq t, \arg\max(f^j(\boldsymbol{x}_i)) = \arg\max(f^k(\boldsymbol{x}_i)), \tag{6}$$

$$\omega_{i3} = \forall 1 \leq j \leq t, \max(f^j(\boldsymbol{x}_i)) \geq \lambda, \tag{7}$$

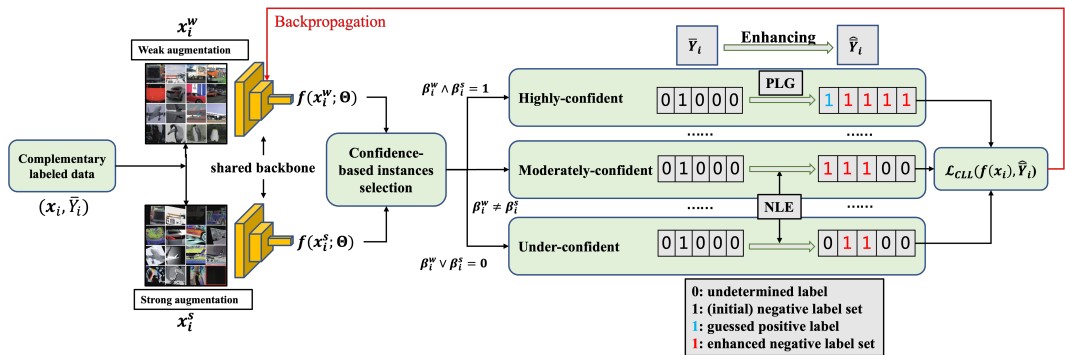

Figure 1: The overall framework of PLNL. We employs a two-view network (shared backbone) to extract features and compute confidences for weak and strong augmentations of one instance respectively. After the selection of highly-confident, moderately-confident and under-confident set, We employ PLG on highly-confident instances and NLE for the rest. The loss is computed on the enhanced labels of both views and the model updates through backpropagation.

where $\omega_{i1}, \omega_{i2}, \omega_{i3}$ are boolean variables which indicate whether the corresponding criterion is satisfied. $\omega_{i1}$ ensures that the label corresponding to the max prediction confidence does not fall on the complementary label set, which excludes the complementary labels from being selected as positive label. $\omega_{i2}$ ensures that the max prediction confidence be stable and show no sign of fluctuations over the past $t$ epochs. $\omega_{i3}$ ensures the max prediction confidence should be higher than a threshold $\lambda$. Note that $\lambda$ can be either a man-made threshold or a self-adaptive one, which will be discussed in detail later.

**Warm up.** Before selecting, we warm up the model using the entire training set. The goal of this stage is to reduce the classification risk and obtain some historical prediction confidence to construct the memory bank since we have no historical information at initial epoch. In this paper, we use SCL-LOG algorithm (Chou et al., 2020) to warm up models for 20 epochs.

**Instance-aware self-adaptive threshold.** The threshold in criterion $\omega_{i3}$, as is mentioned above, can be a fixed high threshold (like 0.95). However, a single global threshold does not consider the fitting difficulties of different instances, i.e., hard instances and easy instances. This will result in very few samples being selected in the early training stages as well as confirmation bias. Therefore, to comprehensively consider historical information, we design an instance-aware self-adaptive threshold for each instance at each epoch $t$ as:

$$\lambda(t) = \alpha\lambda(t-1) + (1-\alpha)f(t), \lambda(0) = \frac{1}{K}, \tag{8}$$

where $K$ is the number of classes, $f(t) = \max f(\boldsymbol{x})$ and $\alpha$ is the ratio which controls the threshold stability.

The threshold is initialized at a low value $\frac{1}{K}$, which will take more data into account and helps speed up convergence in the early stages. As the prediction confidence increases, the threshold grows higher to filter out wrong pseudo labels to alleviate the confirmation bias. Note that we compute $\lambda^w(t)$ and $\lambda^s(t)$ for two different views respectively according to Eq. (8). We use the momentum average confidence of each instance, computed based on all previous epochs. In this way, the threshold comprehensively considers historical information and remains stable and trustworthy.

**Subset Selection.** For the two-view network, we perform two independent verifications. Let $\beta_i^w = \omega_{i1} \wedge \omega_{i2} \wedge \omega_{i3}$ be the indicator of satisfying the criteria. Thus, $\beta_i^w$ and $\beta_i^s$ indicate whether the weak and strong views meet the criteria respectively. For an instance, if both views meet the criteria, we select it to the highly-confident subset, i.e.,

$$\mathcal{H} = \{\boldsymbol{x}_i | \beta_i^w \wedge \beta_i^s = 1\}. \tag{9}$$

It means the prediction confidences of both views are stable and high, thus we consider them to be highly-confident. The size of $\mathcal{H}$ is denoted as $N_h$.

Similarly, the moderately-confident subset consists of instances only one augmented version of which meet the criteria, i.e.,

$$\mathcal{M} = \{\boldsymbol{x}_i | \beta_i^w \neq \beta_i^s\}. \tag{10}$$

It means only one view's prediction is trustworthy, the other is not. which shows that the model is moderately-confident about its prediction. The size of $\mathcal{M}$ is denoted as $N_m$.

Finally, the under-confident subset consists of the rest of the instances, i.e.,

$$\mathcal{U} = \{\boldsymbol{x}_i | \beta_i^w \vee \beta_i^s = 0\}. \tag{11}$$

It means the prediction confidences of both views do not meet the designed criteria, these instances are considered under-confident. The size of $\mathcal{U}$ is denoted as $N_u$.

After obtaining $\mathcal{H}$, $\mathcal{M}$ and $\mathcal{U}$, we design different techniques to better utilize these different types of training instances.

### 3.2  POSITIVE LABEL GUESSING

For highly-confident set $\mathcal{H}$, we consider the label with the max prediction confidence as its positive label. **Conversely, all remaining labels are considered complementary labels.** Let $\widehat{\bar{Y}}_i$ be the enhanced negative label set for instance $\boldsymbol{x}_i$, we have:

$$\widehat{\bar{Y}}_i = \{c | c \in Y_i, c \neq \hat{y}_i\}, \tag{12}$$

where $\hat{y}_i$ is the guessed positive label and $Y_i = \{1, 2, \ldots, K\}$ is the full label set.

For highly-confident set $\mathcal{H}$, we compute the CLL loss on the negative labels for both views:

$$\mathcal{L}_h = \frac{1}{N} \sum_{i=1}^{N_h} \bar{\mathcal{L}}_{CLL}(f(\boldsymbol{x}_i^w), \widehat{\bar{Y}}_i) + \bar{\mathcal{L}}_{CLL}(f(\boldsymbol{x}_i^s), \widehat{\bar{Y}}_i), \tag{13}$$

where $f(\boldsymbol{x}_i^w)$ and $f(\boldsymbol{x}_i^s)$ denote model outputs of weak augmentation and strong augmentation respectively.

### 3.3  NEGATIVE LABEL ENHANCEMENT

For the moderately-confident set $\mathcal{M}$ and the under-confident set $\mathcal{U}$, guessing the positive labels directly might lead to much more errors due to their relatively lower confidence. Therefore, we employ a different strategy for these instances, called negative label enhancement (NLE).

The rationale of NLE is that more negative labels will bring in additional supervision information for better training. However, whether the enhanced negative labels are correct remains a question. Intuitively, randomly enhancing negative labels will bring in a large number of labeling errors. To better enhance the reliability of NLE, we further exploit information in the output space and design the following solution.

**Calculation of $k$ Nearest Neighbor ($k$-NN) instances.**  For instance $\boldsymbol{x}_i$ and its model output prediction confidence $y_i$, we can compute its $k$-NN instances in the output space. It is safe to assume that nearby instances in the output space should have the same positive label with a high probability, while their original complementary label sets are likely to vary. The formal definition of this assumption is as follows:

**Assumption 1.** $\forall (\boldsymbol{x}_i, \bar{Y}_i) \in \mathcal{D}$ and its $k$-NN instances $(\boldsymbol{x}_i^{(j)}, \bar{Y}_i^{(j)})$, the positive label $y_i$ exists in its $k$-NN instances' complementary label set $\bar{Y}_i^{(j)}$ with probability no more than $\alpha_k$, any negative label $y_i' \neq y_i$ exist in its $k$-NN instances' complementary label set $\bar{Y}_i^{(j)}$ with probability no less than $\beta_k$.

This assumption describes the intrinsic characteristics of CLL in the output space, which can be interpreted in two aspects. First, similarity in the input space will be mapped to similarity in the output space, which has been widely utilized for tackling representation learning problems (He et al., 2020). Second, instances of the same category are likely to be labeled with complementary labels of different categories, which is key to enhancing the negative labels.

$k$**-NN label frequency.** For instance $\boldsymbol{x}_i$, we propose to calculate the times a negative label appears in its $k$-NN instances' complementary label set and then enhance top-$\tau_i$ frequent ones, that is, add them to the complementary label set of $\boldsymbol{x}_i$. We define the $j$-th $k$-NN label frequency of $\boldsymbol{x}_i$ as follows:

$$\boldsymbol{F}_{ij} = \sum\nolimits_{v=1}^{k} \mathbb{I}(j \in \bar{Y}_i^{(v)}),\qquad(14)$$

where $\bar{Y}_i^{(v)}$ denotes the complementary label set of the $v$-th nearest instance of $\boldsymbol{x}_i$.

**Negative label enhancement.** We enhance the complementary label set $\bar{Y}_i$ by adding additional labels with top-$\tau_i$ label frequency. For each instance $\boldsymbol{x}_i$ in $\mathcal{M}$ and $\mathcal{U}$, the enhanced complementary (negative) label set $\widehat{\bar{Y}}_i$ is calculated by:

$$\widehat{\bar{Y}}_i = \{c | c \in \bar{Y}_i \vee c \in \text{top-}\tau_i\text{-max}_{j \in Y_i}(\boldsymbol{F}_{ij})\}.\qquad(15)$$

However, the prediction confidence of the under-confident is more unreliable than that of the moderately-confident. Therefore, we should be more conservative when enhancing these instances as the $k$-NN information may be more unreliable. In our work, we set $\tau_i = \lceil \frac{K-s_i}{10} \rceil$ for $\mathcal{U}$ where $s_i$ is the size of $\bar{Y}_i$. For $\mathcal{M}$, we set $\tau_i = (1 + \frac{e}{E_{max}})\lceil \frac{K-s_i}{10} \rceil$ where $e$ is current epoch, $E_{max}$ is total epochs. This provides a linear growing strategy for the moderately-confident because the model's output becomes increasingly accurate as the training progresses. The ablation study on this strategy is provided in section G.6.

For moderately-confident set $\mathcal{M}$ and under-confident set $\mathcal{U}$, we compute the CLL loss on the negative labels for both views:

$$\mathcal{L}_{m,u} = \frac{1}{N} \sum_{i=1}^{N_m+N_u} \bar{\mathcal{L}}_{CLL}(f(\boldsymbol{x}_i^w), \widehat{\bar{Y}}_i) + \bar{\mathcal{L}}_{CLL}(f(\boldsymbol{x}_i^s), \widehat{\bar{Y}}_i),\qquad(16)$$

where $f(\boldsymbol{x}_i^w)$ and $f(\boldsymbol{x}_i^s)$ denote model outputs of weak augmentation and strong augmentation respectively.

## 4 A UNIFIED FRAMEWORK FOR PSEUDO-LABELING-BASED METHODS

Pseudo-labeling, which has been widely used in the recent semi-supervised learning (SSL) methods, is employed by giving unlabeled instances pseudo labels and train them in a supervised way. PLNL is an extension of pseudo-labeling. We not only consider pseudo-labeling of highly-confident instances, but also consider enhancing the negative label set of untrustworthy instances. In this way, we actually recover more supervised information than only leveraging pseudo-labeling and further boost the classification performance.

In this section, we construct a unified framework where PLG and NLE are viewed from the perspective of negative label recovery. Let $\hat{y}_i$ be the pseudo-label of $\boldsymbol{x}_i$. Let $Y_i$ be the full label space. Let $\widehat{\bar{Y}}_i$ be the reconstructed (PLNL) or imposed (pseudo-labeling) negative label set for $\boldsymbol{x}_i$.

For PLNL, PLG is equivalent to reconstructing a negative label set $\widehat{\bar{Y}}_i = \{c | c \in Y_i, c \neq \hat{y}_i\}$ of size $K - 1$, in which only the guessed positive label does not belong. NLE is equivalent to reconstructing a negative label set of size $s_i + \tau_i$, where we add $\tau_i$ negative labels to the original negative label set of size $s_i$.

Similarly, for pseudo-labeling methods, let the pseudo label for instance $\boldsymbol{x}_i$ be $\hat{y}_i$ as well. The process of pseudo-label highly-confident instances is also equivalent to imposing a negative label set $\widehat{\bar{Y}}_i = \{c | c \in Y_i, c \neq \hat{y}_i\}$ of size $K - 1$ as additional supervised information.

In this paper, we propose two metrics for evaluation of pseudo-labeling-based methods. Firstly, we define selected ratio $\eta$:

$$\eta = \frac{N_h}{N}.\qquad(17)$$

Obviously, $\eta$ evaluates the ratio of highly-confident instances selected for pseudo-labeling methods.

Furthermore, we define average size of enhanced negative label set $\bar{s}$:

$$\bar{s} = \frac{\sum_{i=1}^{N} |\widehat{\bar{Y}}_i|}{N}. \tag{18}$$

In section 6, we empirically show that PLNL achieves lower error rate $\epsilon$, higher selection ratio $\eta$ and obviously larger size of negative label set $\bar{s}$ compared with pseudo-labeling method Fixmatch (Sohn et al., 2020) and Freematch (Wang et al., 2023b).

## 5 THEORETICAL ANALYSIS

### 5.1 GENERALIZATION BOUND

For simplification, we only consider one view network here, which has no influence on the deduction of generalization error bound. Our goal is to learn a classification model $f(\boldsymbol{x}; \Theta)$ by minimizing the empirical risk $\widehat{R}'(f)$ acquired from data with enhanced negative labels:

$$\widehat{R}'(f) = \frac{1}{N} \sum_{i=1}^{N} \bar{\mathcal{L}}_{CLL}(f(\boldsymbol{x}_i), \widehat{\bar{Y}}_i), \tag{19}$$

where $\widehat{\bar{Y}}_i$ denotes the enhanced negative label set of $\boldsymbol{x}_i$.

Let the CLL loss function be $\mathcal{L}_{CLL}(f(\boldsymbol{x}), \widehat{\bar{Y}}_i) = \sum_{y \notin \widehat{\bar{Y}}_i} (1/(K - |\widehat{\bar{Y}}_i|)) \ell(f(\boldsymbol{x}), y)$ where $|\widehat{\bar{Y}}_i|$ is the size of the enhanced negative label set. Let $\bar{s} = \frac{\sum_{i=1}^{N} |\widehat{\bar{Y}}_i|}{N}$ be the average size of enhanced negative label set. Let $\epsilon_1 = \sum_{i=1}^{N_h} \frac{\mathbb{I}(y_i \in \widehat{\bar{Y}}_i)}{N_h}$ be the error rate of PLG. Let $\epsilon_2 = \sum_{i=1}^{N_m + N_u} \frac{\mathbb{I}(y_i \in \widehat{\bar{Y}}_i)}{N_m + N_u}$ be the error rate of NLE. The actual pseudo-labeling error rate $\epsilon = \frac{\sum_{i=1}^{N} \mathbb{I}(y_i \in \widehat{\bar{Y}}_i)}{N} = \frac{N_h}{N}\epsilon_1 + \frac{N_m + N_u}{N}\epsilon_2 = \eta\epsilon_1 + (1-\eta)\epsilon_2$. Moreover, $\ell(f(\boldsymbol{x}), y)$ is $\rho$-Lipschitz *w.r.t.* $f(\boldsymbol{x})$ where $\rho$ can be any Lipschitz constant (not necessarily the best). Let $\Re_N(\mathcal{F})$ be the expected Rademacher complexity (Mohri et al., 2012) of $\mathcal{F}$ with $N$ training instances. Let $\hat{f} = \text{argmin}_{f \in \mathcal{F}} \widehat{R}'(f)$ be the empirical risk minimizer, where $\mathcal{F}$ is a function class, and $f^* = \text{argmin}_{f \in \mathcal{F}} R(f)$ be the true risk minimizer. We derive the following theorem, which provides a generalization error bound for the proposed method.

**Theorem 1.** *Suppose that $\ell(f(\boldsymbol{x}), y)$ is bounded by $B$. For pseudo-labeling error rate $\epsilon \in (0, 1)$, for any $\delta > 0$, with probability at least $1 - \delta$, we have*

$$R(\hat{f}) - R(f^*) \leq 2(1 - \frac{1 - \epsilon}{K - \bar{s}})B + 4\rho K \Re_N(\mathcal{F}) + 2KB\sqrt{\frac{\log\frac{2}{\delta}}{2N}}. \tag{20}$$

***Remark.*** Detailed proofs are provided in Appendix B. Theorem 1 shows that as $N \to \infty$, $\epsilon_1 \to 0$, $\epsilon_2 \to 0$, the empirical risk minimizer converges to the true risk minimizer with high probability. It can be observed from Eq. (20) that the generalization bound is influenced by five factors: the number of categories $K$, the average size of enhanced negative label set $\bar{s}$ and two error rates. This is consistent with the intuition that more categories and less complementary labels will make the CLL problem harder. **In a nutshell, smaller PLNL pseudo-labeling error rates $\epsilon_1, \epsilon_2$ and larger size of enhanced negative label set $\bar{s}$ will produce better generalization performance.**

### 5.2 ERROR BOUND OF POSITIVE LABEL GUESSING

**Theorem 2.** *Let $y_i$ denote the ground-truth positive label of $\boldsymbol{x}_i$ and $\hat{y}_i$ denote the guessed positive label which might not be true. PLG error rate $\epsilon_1$ is upper bounded by:*

$$\epsilon_1 = \mathbb{P}(y_i \in \widehat{\bar{Y}}_i) \leq \psi(\boldsymbol{X}, \bar{\boldsymbol{Y}})^{(K-1-s_i)}, \tag{21}$$

where $K$ is class number, $s_i$ is the size of $\bar{Y}_i$, $\psi$ is a positive constant approaching zero, which is related to the probabilistic distribution where $(\boldsymbol{x}_i, \bar{Y}_i)$ is sampled. Detailed proofs are provided in Appendix C.

## 5.3 ERROR BOUND OF NEGATIVE LABEL ENHANCEMENT

**Theorem 3.** *Let $y_i$ denote the ground-truth positive label of $\boldsymbol{x}_i$ and $y'$ denote an arbitrary negative label. Let $\boldsymbol{F}_i^{(\tau_i)}$ denote the $\tau_i$-th largest label frequency. Let $p$ denote the probability of the ground-truth positive label $y_i$ appearing in its $k$-NN instance's complementary label set. Let $q$ denote the probability of the label $y'$ appearing in its $k$-NN instance's complementary label set. The NLE error rate $\epsilon_2$ is upper bounded by:*

$$\epsilon_2 = \mathbb{P}(y_i \in \widehat{\widehat{Y}}_i) \leq \sum_{j=1}^{k} \binom{|Y_i|-1}{|Y_i|-\tau_i} I_{\beta_k}(k-j+1,j)^{(|Y_i|-\tau_i)} b_{\alpha_k}(k,j), \tag{22}$$

*where $I_{\beta_k}(k,j) = \int_0^{\beta_k}(1-q)^{k-1}q^{j-1}dq$ denotes the regularized incomplete beta function, $b_{\alpha_k}(k,j) = \binom{k}{j}\alpha_k^j(1-\alpha_k)^{k-j}$ is the probability mass function of a binomial distribution $B(k,\alpha_k)$. Detailed proofs are provided in Appendix D.*

***Remark.*** Theorem 2 and Theorem 3 show that both PLG error rate $\epsilon_1$ and NLE error rate $\epsilon_2$ are upper bounded under mild condition.

Table 1: Comparison of classification accuracies between different methods on four datasets with a single complementary label per instance. The results (mean $\pm$ std) are reported over 3 random trials. The best results are highlighted in bold (The same applies hereinafter).

| Method | STL-10 | SVHN | FMNIST | CIFAR-10 |
|---|---|---|---|---|
| UB-EXP | 28.84±0.54% | 88.93±0.17% | 87.96±0.08% | 62.90±0.06% |
| UB-LOG | 20.41±0.46% | 89.59±0.08% | 87.59±0.14% | 70.28±0.12% |
| SCL-EXP | 31.03±0.61% | 88.66±0.20% | 88.31±0.09% | 72.35±0.10% |
| SCL-LOG | 30.74±0.72% | 89.26±0.24% | 88.03±0.10% | 79.87±0.14% |
| POCR | 34.96±0.32% | 96.65±0.14% | 92.29±0.07% | 94.15±0.09% |
| SELF-CL | 30.87±0.72% | 90.13±0.23% | 84.86±0.10% | 88.95±0.22% |
| ComCo | 32.43±0.28% | 91.41±0.35% | 85.42±0.40% | 89.36±0.76% |
| Ours | **55.25±0.36%** | **97.58±0.18%** | **93.38±0.06%** | **94.78±0.12%** |

Table 2: Comparison of classification accuracies between different methods on five datasets with multiple complementary labels per instance. The results (mean $\pm$ std) are reported over 3 random trials.

| Method | STL-10 | SVHN | FMNIST | CIFAR-10 | CIFAR-100 |
|---|---|---|---|---|---|
| UB-EXP | 60.85±0.12% | 95.23±0.09% | 92.34±0.28% | 91.13±0.23% | 34.43±0.08% |
| UB-LOG | 62.84±0.17% | 94.76±0.07% | 91.84±0.29% | 92.01±0.21% | 52.76±0.15% |
| SCL-EXP | 62.96±0.10% | 95.28±0.14% | 92.20±0.27% | 91.85±0.25% | 47.81±0.09% |
| SCL-LOG | 61.60±0.14% | 94.88±0.16% | 91.51±0.25% | 92.67±0.18% | 49.40±0.19% |
| POCR | 74.51±0.29% | 97.14±0.09% | 94.76±0.26% | 96.09±0.27% | 53.16±0.11% |
| SELF-CL | 69.85±0.20% | 91.58±0.30% | 94.92±0.21% | 92.23±0.16% | 57.65±0.25% |
| ComCo | 73.28±0.19% | 95.41±0.23% | 92.01±0.16% | 91.38±0.73% | 57.88±0.95% |
| Ours | **77.11±0.14%** | **98.13±0.11%** | **95.16±0.13%** | **96.80±0.28%** | **64.33±0.43%** |

## 6 EXPERIMENT

### 6.1 EXPERIMENT SETUP

**Datasets.** We use five commonly used benchmark datasets, STL-10 (Coates et al., 2011), Fashion-MNIST (FMNIST) (Xiao et al., 2017), SVHN (Netzer et al., 2011), CIFAR-10 and CIFAR-100 (Krizhevsky and Hinton, 2009). We conduct experiments by considering both the scenarios of Single CLL (SCLL) and Multiple CLL (MCLL). The datasets generation process are provided in Appendix F.

**Compared methods.** We compare the performance of PLNL with seven state-of-the-art CLL methods, including UB-EXP (Feng et al., 2020), UB-LOG (Feng et al., 2020), SCL-EXP (Chou et al., 2020), SCL-LOG (Chou et al., 2020), POCR (Wang et al., 2021), SELF-CL (Liu et al., 2022) and ComCo (Jiang et al., 2024a) and two state-of-the-art SSL methods, Fixmatch (Sohn et al., 2020) and Freematch (Wang et al., 2023b).

**Implementation.** Implementation details are provided in Appendix F.

## 6.2 MAIN RESULTS

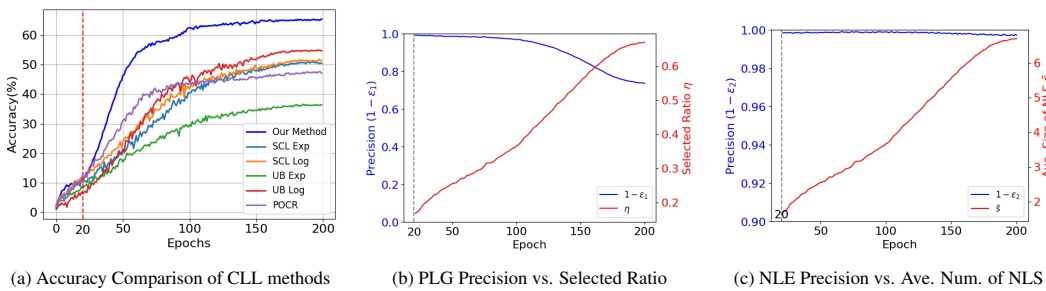

(a) Accuracy Comparison of CLL methods    (b) PLG Precision vs. Selected Ratio    (c) NLE Precision vs. Ave. Num. of NLS

Figure 2: The experiments is conducted on CIFAR-100 with multiple complementary labels (MCLL). (a) The accuracy of PLNL improves tremendously over epochs and achieves the best finally. (b) The precision of PLG decreases slowly, while the selected ratio steadily rises, indicating a growing proportion of selected instances during training. (c) Ave. Size of NLS denotes average number of negative label set $\bar{s}$. The precision of NLE remains relatively stable with a slight decrease, whereas the average size of negative label set increases significantly, showing a steady recovery of negative labels.

**PLNL achieves SOTA results.** As shown in Table 1 and Table 2, PLNL outperforms all the compared method by a significant margin across all datasets. Specifically, on STL-10 dataset, we outperform the previous SOTA by **20.29%** and **1.61%** in both SCLL and MCLL settings. Furthermore, PLNL performs even better in harder scenarios where there is larger label space or less supervised information for each class. We challenge this by showing our results on CIFAR-100 datasets. On CIFAR-100 with Multiple CLs, the improvement is **6.45%** compared to previous SOTA. Fig. 2a further demonstrates that PLNL significantly outperforms the compared ones.

**PLNL pseudo-labeling achieves excellent performance with extremely high precision.** Fig. 2b shows that as the number of epochs increases, PLG will select more and more highly-confident instance, eventually occupying most of the dataset, while the precision only drops slightly in the final stage, which maintains high precision and high selected ratio. Meanwhile, Fig. 2c shows that NLE identifies more and more negative labels with extremely high precision, which maintains above 0.99 throughout training.

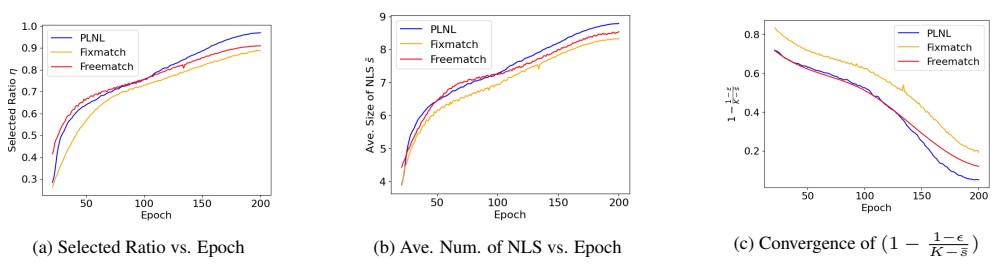

(a) Selected Ratio vs. Epoch    (b) Ave. Num. of NLS vs. Epoch    (c) Convergence of $\left(1 - \frac{1-\epsilon}{K-\bar{s}}\right)$

Figure 3: The experiments is conducted on CIFAR-10 with single complementary labels (SCLL). (a) shows that selected ratio of PLNL trenscends Fixmatch and Freematch significantly. (b) shows that average size of NLS of PLNL is significantly larger due to specially designed technique NLE for enhancing the untrustworthy negative labels. Nearly all negative labels are revealed at the end of training, almost reaching 9 negative labels for each instances in CIFAR-10 (c) indicates that the value of $\left(1 - \frac{1-\epsilon}{K-\bar{s}}\right)$ decreases steadily during training.

**Compared with SOTA semi-supervised learning methods, PLNL performs even better.** From Fig. 3a and Fig. 3b, we can see that PLNL achieves both higher selected ratio and recovered more negative labels compared with Fixmatch and Freematch. PLNL is both accurate and comprehensive in recovering label information. This highlights PLNL's enhanced capacity in leveraging moderately-confident and under-confident instances for label recovery, showcasing both stability and scalability.

**PLNL can reduce the generalization bound and achieves lower generalization bound compared with SSL methods.** Fig. 3c demonstrates that $(1 - \frac{1-\epsilon}{K-\bar{s}})$ decreases as the training progresses. As $(1 - \frac{1-\epsilon}{K-\bar{s}})$ is the variable of the generalization bound derived in Theorem 1, it is safe to conclude that PLNL can continuously and significantly reduce the generalization error.

## 6.3 ABLATION STUDY

**Two-view networks facilitate increased pseudo-labeling precision.** We observe that two-view network significantly boost the performance of PLNL pseudo-labeling, which helps accurately select more highly-confident instances. As shown in Table 3, the $\eta$ increases **8.75%** and **8.65%** respectively on CIFAR-10 and CIFAR-100 with $1 - \epsilon_1$ increases **6.58%** and **9.22%** respectively. We also compare PLNL with one variant: PLNL v1 where we replace the two-view networks in PLNL with a single network in Table 4, which shows that ours outperforms PLNL v1 by a remarkable margin (e.g. **+6.00%** on STL-10).

Table 3: The performance of PLNL with single network on two settings.

| Method | CIFAR-10 SCLL | | CIFAR-100 MCLL | |
|---|---|---|---|---|
| | $\eta$ | $1$-$\epsilon_1$ | $\eta$ | $1$-$\epsilon_1$ |
| Single | 84.65 | 90.21 | 67.43 | 70.62 |
| Two-view | **93.40** | **96.79** | **76.08** | **79.84** |

**All the components contribute to the performance gain.** We explore the effectiveness of our proposed PLG and NLE method on three settings STL-10 (SCLL), CIFAR-10 (SCLL), CIFAR-100 (MCLL). Specifically, we compare PLNL with several variants: (1) PLNL v2 which removes the PLG component; (2) PLNL v3 which removes the NLE component. From Table 4, we observe that PLNL outperforms PLNL v2 remarkably (e.g., **+5.43%** on STL-10), which proves the effectiveness of positive label guessing. We also observe that ours outperforms PLNL v3 (e.g., **+1.19%** on CIFAR-100), which proves the effectiveness of NLE.

Table 4: Classification accuracy of degenerated methods on three settings.

| Method | STL-10 SCLL | CIFAR-10 SCLL | CIFAR-100 MCLL |
|---|---|---|---|
| PLNL | **55.25** | **94.78** | **64.33** |
| PLNL v1 | 49.25 | 93.75 | 63.09 |
| PLNL v2 | 49.82 | 92.01 | 58.94 |
| PLNL v3 | 53.22 | 94.28 | 63.14 |
| POCR | 34.96 | 94.15 | 53.16 |

## 7 CONCLUSION

In this paper, we introduce a novel complementary label learning method that reformulates CLL as an inverse problem to infer the full label information from the output space information. To this end, we split this inverse problem into two subtasks (PLG and NLE). A confidence-based instances selection module is proposed for dataset split: highly-confident, moderately-confident and under-confident. Then we perform PLG for highly-confident instances by assigning pseudo-labels to them. For moderately-confident and under-confident instances, we perform NLE by enhancing their negative label set with different levels and train them with the augmented negative labels iteratively. We theoretically prove that when pseudo-labeling error is limited, we can construct a classifier consistent with that learned by clean full labels. The upper bounds of PLG and NLE error rate are deduced and we empirically show that PLNL can infer both positive and negative labels with a high precision. We conducted extensive experiments which demonstrate that PLNL achieves a new state-of-the-art in CLL. In addition, extensive ablation studies have proved the effectiveness of each component.

ACKNOWLEDGEMENTS

This work was supported by the National Natural Science Foundation of China under Grants U24A20322 and 62302094. This work was also supported by the Big Data Computing Center of Southeast University.

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

## A    THE PSEUDO-CODE OF PLNL

---

**Algorithm 1** Pseudo-code of PLNL.

---

**Input:** Training dataset $\mathcal{D}$, mini-batch size $B$, epochs $E_{max}$, a two-view network with shared parameter $\Theta$ and two memory bank matrices $\boldsymbol{M}^w, \boldsymbol{M}^s$, hyperparameters $t, k, \tau_m, \tau_u$.
 1: Initialize $\boldsymbol{M}^w, \boldsymbol{M}^s, \Theta$ by warming up 20 epochs using SCL-LOG (Chou et al., 2020).
 2: **for** $e = 1, 2, \ldots, E_{max}$ **do**
 3:    Shuffle $\mathcal{D}$ into $\frac{|\mathcal{D}|}{B}$ mini-batches;
 4:    Construct $\boldsymbol{M}^w, \boldsymbol{M}^s$;
 5:    Compute two-view network outputs of each instances;
 6:    Select $\mathcal{H}, \mathcal{M}, \mathcal{U}$ based on criteria in Eq. (5);
 7:    Pseudo-label the highly-confident set $\mathcal{H}$; // PLG
 8:    Enhance negative labels of set $\mathcal{M}, \mathcal{U}$ based on Eq. (14) and Eq. (15); // NLE
 9:    **for** $i = 1, 2, \ldots, \frac{|\mathcal{D}|}{B}$ **do**
10:       Fetch an batch $\tilde{B}_i$ with enhanced negative label set;
11:       Compute the empirical risk $\widehat{R}'(f)$ by Eq. (19);
12:       Update parameters of $\Theta$;
13:    **end for**
14: **end for**
**Output:** parameters of $\Theta$.

---

## B    PROOF OF THEOREM 1

We first derive the uniform deviation bound between $\widehat{R}(f)$ and $R(f)$.

**Lemma 1.** *Suppose that the binary loss function $\ell(f(\boldsymbol{x}), y)$ is $\rho$-Lipschitz continuous w.r.t. $f(\boldsymbol{x})$. For any $\delta > 0$, with probability at least $1 - \delta$, we have*

$$\left| R(f) - \widehat{R}(f) \right| \le 2\rho K \mathfrak{R}_N(\mathcal{F}) + KB \sqrt{\frac{\log \frac{2}{\delta}}{2N}}. \tag{23}$$

*Proof.* We firstly define the Rademacher complexity of $\mathcal{L}$ and $\mathcal{F}$ with $N$ training instances as follows:

$$\mathfrak{R}_N(\mathcal{L} \circ \mathcal{F})$$
$$= \mathbb{E}_{\boldsymbol{x}, \boldsymbol{y}, \boldsymbol{\sigma}} \left[ \sup_{f \in \mathcal{F}} \sum_{i=1}^{N} \sigma_i \mathcal{L}_{CLL}(f(\boldsymbol{x}_i), \widehat{\overline{Y}}_i),) \right]. \tag{24}$$

Considering that $\mathcal{L}_{CLL}(f(\boldsymbol{x}), \widehat{\overline{Y}}_i) = \sum_{y \notin \widehat{\overline{Y}}_i} \ell(f(\boldsymbol{x}), y)$, we have

$$\mathfrak{R}_N(\mathcal{L} \circ \mathcal{F}) \le K \mathfrak{R}_N(\ell \circ \mathcal{F}),$$
$$\le \rho K \mathfrak{R}_N(\mathcal{F}), \tag{25}$$

where the second line is based on Lipschitz continuity of $\ell(f(\boldsymbol{x}), y)$.

Suppose an instance $(\boldsymbol{x}_i, y_i)$ is replaced by another arbitrary instance $(\boldsymbol{x}_i^{'}, y_i^{'})$, this leads to a change of $\sup_{f \in \mathcal{F}} R(f) - \widehat{R}(f)$ no greater than $\frac{KB}{N}$ due to the fact that $\ell$ is bounded by $B$. According to McDiarmid's inequality (Mohri et al., 2012), for any $\delta > 0$, with probability at least $1 - \frac{\delta}{2}$, we have

$$\sup_{f \in \mathcal{F}} R(f) - \widehat{R}(f) \le \mathbb{E} \left[ \sup_{f \in \mathcal{F}} R(f) - \widehat{R}(f) \right] + KB \sqrt{\frac{\log \frac{2}{\delta}}{2N}}. \tag{26}$$

In addition, it is routine (Mohri et al., 2012) that

$$\mathbb{E} \left[ \sup_{f \in \mathcal{F}} R(f) - \widehat{R}(f) \right] \le 2 \bar{\mathfrak{R}}_N(\mathcal{F}). \tag{27}$$

By combining Eq. (26) and Eq. (27), and futher taking the other direction of $\sup_{f \in \mathcal{F}} \widehat{R}(f) - R(f)$ into account, with probability at least $1 - \delta$, we have

$$\sup_{f \in \mathcal{F}} \left| R(f) - \widehat{R}(f) \right| \leq 2\rho K \mathfrak{R}_N(\mathcal{F}) + KB\sqrt{\frac{\log \frac{2}{\delta}}{2N}}, \qquad (28)$$

which concludes the proof.

Due to inevitable error made in PLG and NLE. The problem of CLL actually has been translated into a noisy CLL problem where the true label might be mislabeled as complementary label (Ishiguro et al., 2022). Let $\epsilon_1$ be the error rate of PLG and $\epsilon_2$ be the error rate of NLE. Since NLE error rate $\epsilon_1 = \sum_{i=1}^{N_h} \frac{\mathbb{I}(y_i \in \widehat{Y}_i)}{N_h}$ and NLE error rate $\epsilon_2 = \sum_{i=1}^{N_m+N_u} \frac{\mathbb{I}(y_i \in \widehat{Y}_i)}{N_m+N_u}$, the actual noise rate $\epsilon = \frac{\sum_{i=1}^{N} \mathbb{I}(y_i \in \widehat{Y}_i)}{N}$ can be calculated as $\epsilon = \frac{N_h}{N}\epsilon_1 + \frac{N_m+N_u}{N}\epsilon_2 = \eta\epsilon_1 + (1-\eta)\epsilon_2$. We further bound the difference between $\widehat{R}(f)$ and $\hat{R}'(f)$.

**Lemma 2.** *Suppose that the binary loss function $\ell$ is bounded by $B$. For some noise rate $\epsilon \in (0,1)$ and average complementary label size $\bar{s}$ for any $f \in \mathcal{F}$, we have*

$$\left| \widehat{R}'(f) - \widehat{R}(f) \right| \leq (1 - \frac{1-\epsilon}{K-\bar{s}})B. \qquad (29)$$

We firstly proved the upper bound of the $\widehat{R}'(f)$:

$$\widehat{R}'(f) = \frac{1}{N} \sum_{i=1}^{N} \bar{\mathcal{L}}_{CLL}(f(\boldsymbol{x}_i), \widehat{Y}_i),$$

$$= \frac{1}{N} \sum_{i=1}^{N} \ell(f(\boldsymbol{x}_i), y_i) + \frac{1}{N} \sum_{i=1}^{N} \mathbb{I}(y_i \notin \widehat{Y}_i) \left[ \sum_{c \notin \widehat{Y}_i, c \neq y_i} \frac{1}{K - |\widehat{Y}_i|} \ell(f(\boldsymbol{x}_i), c) - \frac{K - |\widehat{Y}_i| - 1}{K - |\widehat{Y}_i|} \ell(f(\boldsymbol{x}_i), y_i) \right]$$

$$+ \frac{1}{N} \sum_{i=1}^{N} \mathbb{I}(y_i \in \widehat{Y}_i) \left[ \sum_{c \notin \widehat{Y}_i} \frac{1}{K - |\widehat{Y}_i|} \ell(f(\boldsymbol{x}_i), c) - \ell(f(\boldsymbol{x}_i), y_i) \right],$$

$$\leq \widehat{R}(f) + \frac{1}{N} \sum_{i=1}^{N} \mathbb{I}(y_i \notin \widehat{Y}_i) \sum_{c \notin \widehat{Y}_i, c \neq y_i} \frac{1}{K - \bar{s}} \ell(f(\boldsymbol{x}_i), c) + \frac{1}{N} \sum_{i=1}^{N} \mathbb{I}(y_i \in \widehat{Y}_i) \sum_{c \notin \widehat{Y}_i} \frac{1}{K - \bar{s}} \ell(f(\boldsymbol{x}_i), c),$$

$$\leq \widehat{R}(f) + (1 - \epsilon) \frac{K - \bar{s} - 1}{K - \bar{s}} B + \epsilon B, \qquad (30)$$

where the second line holds based on $\epsilon = \frac{\sum_{i=1}^{N} \mathbb{I}(y_i \in \widehat{Y}_i)}{N}$ and Jensen's inequality (Abramovich et al., 2004). Detailed proof of Eq. (30) is provided in Appendix I. We can prove the lower bound in a similar way:

$$\widehat{R}'(f) = \frac{1}{N} \sum_{i=1}^{N} \ell(f(\boldsymbol{x}_i), y_i) + \frac{1}{N} \sum_{i=1}^{N} \mathbb{I}(y_i \notin \widehat{Y}_i) \left[ \sum_{c \notin \widehat{Y}_i, c \neq y_i} \frac{1}{K - |\widehat{Y}_i|} \ell(f(\boldsymbol{x}_i), c) - \frac{K - |\widehat{Y}_i| - 1}{K - |\widehat{Y}_i|} \ell(f(\boldsymbol{x}_i), y_i) \right]$$

$$+ \frac{1}{N} \sum_{i=1}^{N} \mathbb{I}(y_i \in \widehat{Y}_i) \left[ \sum_{c \notin \widehat{Y}_i} \frac{1}{K - |\widehat{Y}_i|} \ell(f(\boldsymbol{x}_i), c) - \ell(f(\boldsymbol{x}_i), y_i) \right],$$

$$\geq \widehat{R}(f) - \frac{1}{N} \sum_{i=1}^{N} \mathbb{I}(y_i \notin \widehat{Y}_i) \sum_{c \notin \widehat{Y}_i, c \neq y_i} \frac{1}{K - \bar{s}} \ell(f(\boldsymbol{x}_i), c) - \frac{1}{N} \sum_{i=1}^{N} \mathbb{I}(y_i \in \widehat{Y}_i) \sum_{c \notin \widehat{Y}_i} \frac{1}{K - \bar{s}} \ell(f(\boldsymbol{x}_i), c),$$

$$\geq \widehat{R}(f) - (1 - \epsilon) \frac{K - \bar{s} - 1}{K - \bar{s}} B - \epsilon B, \qquad (31)$$

By combining these two sides, we have:

$$\left| \widehat{R}'_h(f) - \widehat{R}_h(f) \right| \leq 1 - \frac{1-\epsilon}{K-\bar{s}}B, \qquad (32)$$

which concludes the proof.

Then, for any $\delta > 0$, with probability at least $1 - \delta$, we have

$$R(\hat{f})$$

$$\leq \widehat{R}(\hat{f}) + 2\rho K \mathfrak{R}_N(\mathcal{F}) + KB\sqrt{\frac{\log \frac{2}{\delta}}{2N}},$$

$$\leq \widehat{R}'(\hat{f}) + (1 - \frac{1-\epsilon}{K-\bar{s}})B + 2\rho K \mathfrak{R}_N(\mathcal{F}) + KB\sqrt{\frac{\log \frac{2}{\delta}}{2N}},$$

$$\leq \widehat{R}'(f^*) + (1 - \frac{1-\epsilon}{K-\bar{s}})B + 2\rho K \mathfrak{R}_N(\mathcal{F}) + KB\sqrt{\frac{\log \frac{2}{\delta}}{2N}},$$

$$\leq \widehat{R}(f^*) + 2(1 - \frac{1-\epsilon}{K-\bar{s}})B + 2\rho K \mathfrak{R}_N(\mathcal{F}) + KB\sqrt{\frac{\log \frac{2}{\delta}}{2N}},$$

$$\leq R(f^*) + 2(1 - \frac{1-\epsilon}{K-\bar{s}})B + 4\rho K \mathfrak{R}_N(\mathcal{F}) + 2KB\sqrt{\frac{\log \frac{2}{\delta}}{2N}}, \tag{33}$$

where the first and fifth line are based on Lemma 1, the second and fourth line are based on Lemma 2, the third line is based on the definition of the empirical risk minimizer $\hat{f} = \operatorname{argmin}_{f \in \mathcal{F}} \widehat{R}(f)$ which means any other $f \neq \hat{f}$ would lead to a larger risk of $\widehat{R}(f)$.

## C  PROOF OF THEOREM 2

Motivated by the formulation of partially labeled data learning in (Gong et al., 2023), here we derive the framework for CLL and PLG from a probabilistic view. Assume that the full label information can be expressed as $Y = \{\bar{Y}, \widetilde{Y}\}$ where $\bar{Y}$ is the complementary label set and $\widetilde{Y}$ is a set of the rest labels (containing the true label). Assume that training dataset $D = \{(\boldsymbol{x}_i, \bar{Y}_i)\}_{i=1}^N$ is drawn i.i.d. $N$ times from a probability distribution $P$, where $\boldsymbol{x}$ is drawn from underlying random variables $\boldsymbol{X} = [\boldsymbol{X}_1, \boldsymbol{X}_2, \dots, \boldsymbol{X}_d]^T$ and $Y$ is drawn from random variables $\boldsymbol{Y} = [\boldsymbol{Y}_1, \boldsymbol{Y}_2, \dots, \boldsymbol{Y}_K]^T$ ($d$ denotes feature dimension and $K$ denotes the number of full labels). And we assume that $Q$ is a hypothetical predictive model with parameters $\theta$, where $\theta$ is the model parameter used for prediction. PLG aims to identify the ground-truth label from $\widetilde{Y}$, which can be implemented by maximizing the conditional likelihood of training dataset with respect to parameters $\theta$. The conditional log-likelihood given all training examples can be expressed as follows.

$$\ell(\theta|D) = \frac{1}{N}\sum_{i=1}^{N} \log Q(\widetilde{Y}|\boldsymbol{x}_i, \theta). \tag{34}$$

By multiplying and dividing classifier Q by the true distribution of identified ground-truth labels given features $P(y|\boldsymbol{x})$, we can formulate Eq. (34) as follows.

$$\ell(\theta|D) = \frac{1}{N}\sum_{i=1}^{N} \log \frac{Q(\widetilde{Y}_i|\boldsymbol{x}_i, \theta)}{P(\widetilde{Y}_i|\boldsymbol{x}_i)} + \frac{1}{N}\sum_{i=1}^{N} \log P(\widetilde{Y}_i|\boldsymbol{x_i}). \tag{35}$$

By multiplying and dividing the probability $P(Y|\boldsymbol{x})$ to the second term, we can formulate Eq. (34) as follows.

$$\ell(\theta|D) = \frac{1}{N}\sum_{i=1}^{N} \log \frac{Q(\widetilde{Y}_i|\boldsymbol{x}_i, \theta)}{P(\widetilde{Y}_i|\boldsymbol{x_i})} + \frac{1}{N}\sum_{i=1}^{N} \log \frac{P(\widetilde{Y}_i|\boldsymbol{x_i})}{P(Y_i|\boldsymbol{x_i})} + \frac{1}{N}\sum_{i=1}^{N} \log P(Y_i|\boldsymbol{x_i}). \tag{36}$$

We use $\mathbb{E}_{(\boldsymbol{X}, \boldsymbol{Y})}$ operator to calculate the expectation of the random variables $(\boldsymbol{X}, \boldsymbol{Y})$, meaning $n \to \infty$.

$$\ell(\theta|D) = \mathbb{E}_{(\boldsymbol{X}, \boldsymbol{Y})}\left\{\log \frac{Q(\widetilde{Y}|\boldsymbol{x}, \theta)}{P(\widetilde{Y}|\boldsymbol{x})}\right\} + \mathbb{E}_{(\boldsymbol{X}, \boldsymbol{Y})}\left\{\log \frac{P(\widetilde{Y}|\boldsymbol{x})}{P(Y|\boldsymbol{x})}\right\} + \mathbb{E}_{(\boldsymbol{X}, \boldsymbol{Y})}\left\{\log P(Y|\boldsymbol{x})\right\}. \tag{37}$$

Recall that we assume that the full label information $Y = \{\bar{Y}, \widetilde{Y}\}$. Then the second term of Eq. (37) can be developed as follows.

$$
\begin{aligned}
\mathbb{E}_{(\boldsymbol{X},\boldsymbol{Y})}\left\{\log\frac{P(\widetilde{Y}|\boldsymbol{x})}{P(Y|\boldsymbol{x})}\right\} &= -\mathbb{E}_{(\boldsymbol{X},\boldsymbol{Y})}\left\{\log\frac{P(Y|\boldsymbol{x})}{P(\widetilde{Y}|\boldsymbol{x})}\right\}, \\
&= -\sum_{(\boldsymbol{x},Y)} P(\boldsymbol{x},Y)\log\frac{P(\widetilde{Y},\bar{Y}|\boldsymbol{x})}{P(\widetilde{Y}|\boldsymbol{x})}, \\
&= -\sum_{(\boldsymbol{x},Y)} P(\boldsymbol{x},Y)\log\frac{P(\widetilde{Y},\bar{Y},\boldsymbol{x})}{P(\widetilde{Y},\boldsymbol{x})}, \\
&= -\sum_{(\boldsymbol{x},Y)} P(\boldsymbol{x},Y)\log\frac{P(\bar{Y},\boldsymbol{x}|\widetilde{Y})}{P(\boldsymbol{x}|\widetilde{Y})}, \\
&= -\sum_{(\boldsymbol{x},Y)} P(\boldsymbol{x},Y)\log\frac{P(\bar{Y},\boldsymbol{x}|\widetilde{Y})}{P(\boldsymbol{x}|\widetilde{Y})}\frac{P(\bar{Y}|\widetilde{Y})}{P(\bar{Y}|\widetilde{Y})}, \\
&= -\sum_{(\boldsymbol{x},Y)} P(\boldsymbol{x},Y)\log\frac{P(\bar{Y},\boldsymbol{x}|\widetilde{Y})}{P(\boldsymbol{x}|\widetilde{Y})P(\bar{Y}|\widetilde{Y})} - \sum_{(\boldsymbol{x},Y)} P(\boldsymbol{x},Y)\log P(\bar{Y}|\widetilde{Y}), \\
&= -\sum_{(\boldsymbol{x},Y)} P(\boldsymbol{x},Y)\log\frac{P(\bar{Y},\boldsymbol{x}|y)}{P(\boldsymbol{x}|y)P(\bar{Y}|\widetilde{Y})} - \sum_{(\bar{Y},\widetilde{Y})} P(\bar{Y},\widetilde{Y})\log P(\bar{Y}|\widetilde{Y}), \\
&= -I(\bar{Y},\boldsymbol{X}|\widetilde{Y}) + H(\bar{Y}|\widetilde{Y}), \\
&= -I(\bar{Y},\boldsymbol{X}|\widetilde{Y}). \qquad\qquad (38)
\end{aligned}
$$

The last equality holds because the conditional entropy $H(\bar{Y}|\widetilde{Y}) = 0$. This is because in CLL, once $\bar{Y}$ is known, then $\widetilde{Y}$ known, meanwhile, the uncertainty remaining in $\widetilde{Y}$ is zero, i.e., $H(\bar{Y}|\widetilde{Y}) = 0$. By combining Eq. (38) and Eq. (37), we have the objective function as follows.

$$
\ell(\theta|D) = -\mathbb{E}_{(\boldsymbol{X},\boldsymbol{Y})}\left\{\log\frac{P(\widetilde{Y}|\boldsymbol{x})}{Q(\widetilde{Y}|\boldsymbol{x},\theta)}\right\} - I(\bar{Y},\boldsymbol{X}|\widetilde{Y}) - H(\boldsymbol{Y}|\boldsymbol{X}). \qquad (39)
$$

The first term is a log likelihood ratio between the true and the predicted ground-truth label distributions given features. The value of this term depends on how well the model $Q$ can approximate $P$. The second term is the conditional mutual information between the complementary labels and the features, given the guessed positive label. The last term is a constant independent of parameters.

Then, we discuss the mild assumption under which PLG method is effective for CLL.

**Assumption 2.** *Let $\boldsymbol{X}$ denote the random variables where $\boldsymbol{x}$ is drawn from. Let $I_y(\bar{Y},\boldsymbol{X}|\widetilde{Y})$ denote the conditional mutual information between the complementary labels and the features, given the ground-truth label $y$ and $I_{y'}(\bar{Y},\boldsymbol{X}|\widetilde{Y})$ denote the conditional mutual information between the complementary labels and the features, given any non-complementary label $y'$. Then, with probability no more than $\psi(\boldsymbol{X},\bar{Y})$, we have*

$$
I_y(\bar{Y},\boldsymbol{X}|\widetilde{Y}) \le I_{y'}(\bar{Y},\boldsymbol{X}|\widetilde{Y}). \qquad (40)
$$

**Remark.** Generally, the value of $I_{\hat{y}}(\bar{Y},\boldsymbol{X}|\widetilde{Y})$ will increase as the guessed positive label $\hat{y}$ explains more about features $\boldsymbol{X}$. $I_{\hat{y}}(\bar{Y},\boldsymbol{X}|\widetilde{Y})$ is a negative number and its maximum value reaches zero when the rest labels in $\widetilde{Y}$ contain no additional information about $\boldsymbol{X}$, namely $\hat{y} = y$. Note that in the absence of label noise and image noise, $\psi(\boldsymbol{X},\bar{Y})$ is a very small value approaching zero and obviously $\psi(\boldsymbol{X},\bar{Y})$ is relevant to specific probability distribution $\boldsymbol{X}$ and $\boldsymbol{Y}$.

In PLG, the training objective actually is the second term of Eq. (39): $\ell(\theta|D) = -I(\bar{Y},\boldsymbol{X}|\widetilde{Y})$. As a result, the error rate of PLG can be calculated as follows. Suppose $\hat{y}$ is the guessed positive label, and

$y'$ is any non-complementary label.

$$\begin{aligned}
\mathbb{P}(y \in \widehat{\widetilde{Y}}), &= \mathbb{P}(y \neq j, \hat{y} = j) \\
&= \prod_{y' \notin \bar{Y}} P(I_y(\bar{Y}, X|\widetilde{Y}) \leq I_{y'}(\bar{Y}, X|\widetilde{Y})), \\
&\leq \psi(X, \bar{Y})^{(K-1-s)},
\end{aligned} \tag{41}$$

where the second line is because once the true label is classified as a complementary label, it means that the conditional mutual information between the complementary labels and the features under every label in $\widetilde{Y}$ being selected as guessed true label is larger than that under $y$ being selected. The third line is based on Assumption 2. This concludes the proof.

## D    PROOF OF THEOREM 3

It is evident that reliable representation information is of great importance to the performace of $k$-NN based NLE. Motivated by the label distinguishability setting in (He et al., 2024), a mild assumption for CLL datasets are discussed to ensure the reliability of the representation information.

The full label information can be expressed as $Y = \{\bar{Y}, \widetilde{Y}\}$ where $\bar{Y}$ is the complementary label set and $\widetilde{Y}$ is a set of the rest labels (containing the true label). According to Assumption 1, we have the following lemma.

**Lemma 3.** $\forall(x_i, \bar{Y}_i) \in \mathcal{D}$, let $p$ denote the probability of the true label $y_i \in Y_i$ appearing in its $k$-NN instance's complementary label set. Let $q$ denote the probability of each non-complementary negative label $y' \in \widetilde{Y}_i \setminus \{y_i\}$ appearing in its $k$-NN instance's complementary label set. Then, we have

$$p \leq \alpha_k, q \geq \beta_k. \tag{42}$$

Then, we derive the error bound of NLE in a step by step manner.

Let $F_i^{(v)}$ denote the $v$-th largest $k$-NN label frequency of instance $x_i$. Assume that $F_{iy}$ and $F_i^{(v)}$ are sampled from binomial distribution $B(k, \alpha_k)$ and binomial distribution $B(k, q)$ respectively. An NLE error is caused by mistakenly adding $y_i$ into $\widehat{\widetilde{Y}}_i$, which means $y_i$ appears more frequently than at least one label that should have been added into $\widehat{\widetilde{Y}}_i$, namely the $\tau_i$-th frequent one. Based on the above, we have

$$\mathbb{P}(y_i \in \widehat{\widetilde{Y}}_i) = \mathbb{P}(F_i^{(\tau_i)} < F_{iy}), \tag{43}$$

$$= \sum_{j=0}^{k} P(F_i^{(\tau_i)} < F_{iy}|F_{iy} = j)P(F_{iy} = j),$$

$$= \sum_{j=1}^{k} P(F_i^{(\tau_i)} \leq j - 1)P(F_{iy} = j),$$

$$= \sum_{j=1}^{k} \left( \underbrace{P(F_i^{(1)} > j)\dots P(F_i^{(\tau_i-1)} > j)}_{(\tau_i-1)items} \underbrace{P(F_i^{(\tau_i)} < j)\dots P(F_i^{(|Y_i|)} < j)}_{(|Y_i|-\tau_i)items} \right) P(F_{iy} = j),$$

$$\leq \sum_{j=1}^{k} \binom{|Y_i| - 1}{|Y_i| - \tau_i} I_{\beta_k}(k - j + 1, j)^{(|Y_i|-\tau_i)} b_{\alpha_k}(k, j), \tag{44}$$

where $I_{\beta_k}(k, j) = \int_0^{\beta_k}(1 - q)^{k-1}q^{j-1}dq$ denotes the regularized incomplete beta function. $b_{\alpha_k}(k, j) = \binom{k}{j}\alpha_k^j(1 - \alpha_k)^{k-j}$ is the probability mass function of a binomial distribution $B(k, \alpha_k)$.

The fourth line holds because the aim is to constrain that there exists $|Y_i| - \tau_i$ labels whose label frequency is less than the true label. The fifth line holds because the following scaling equation holds.

$$P(\boldsymbol{F}_i^{(\tau_i)} < j) \ldots P(\boldsymbol{F}_i^{(|Y_i|)} < j)$$

$$\leq \binom{|Y_i| - 1}{|Y_i| - \tau_i} \prod_{m=1}^{|Y_i| - \tau_i} q_m^{j-1}(1 - q_m)^{k-j},$$

$$\leq \binom{|Y_i| - 1}{|Y_i| - \tau_i} \left( \int_0^{\beta_k} (1 - q)^{k-1} q^{j-1} dq \right)^{(|Y_i| - \tau_i)}. \tag{45}$$

Below, we numerically show that the upper bounds for NLE error rates are small numbers approaching zero.

- $(|Y_i| - 1)$ is a binomial coefficient, which is a non-negative integer, and in this context, it scales the term.
- $I_{\beta_k}(k - j + 1, j)$ is the regularized incomplete beta function, which lies in the range $[0, 1]$.
- $b_{\alpha_k}(k, j)$ represents the probability mass function of a binomial distribution $B(k, \alpha k)$, also within $[0, 1]$.

Since $I_{\beta_k}(k - j + 1, j)$ is within $[0, 1]$, raising it to a high power, specifically $(|Y_i| - \tau_i)$, which is assumed to be large in practical scenarios, causes this term to approach zero, making the contribution of each term in the sum negligible.

Thus, the right-hand side of the inequality is approximately:

$$\sum_{j=1}^{k} \binom{|Y_i| - 1}{|Y_i| - \tau_i} I_{\beta_k}(k - j + 1, j)^{(|Y_i| - \tau_i)} b_{\alpha_k}(k, j) \approx 0. \tag{46}$$

## E    DETAILS OF MCLL SETTINGS

Ishida *et al.* (Ishida et al., 2017) assumed that $P(\boldsymbol{X}, \bar{\boldsymbol{Y}})$ is expressed as:

$$\bar{P}(\boldsymbol{X}, \bar{\boldsymbol{Y}}) = \frac{1}{c - 1} \sum_{\boldsymbol{Y} \neq \bar{\boldsymbol{Y}}} P(\boldsymbol{X}, \boldsymbol{Y}). \tag{47}$$

This assumption implies that all other labels except the correct label are picked as the complementary label with uniform probabilities. Later, Feng *et al.* (Feng et al., 2020) considered a more general setting where each instance is associated with multiple complementary labels (Multiple CLL). Suppose a Multiple CLL dataset is represented as $\{(\boldsymbol{x}_i, \bar{Y}_i)\}_{i=1}^N$, where $\bar{Y}_i$ is the complementary label set for instance $\boldsymbol{x}_i$. Let us denote the number of the complementary labels by a random variable $s_i$, which is sampled from a distribution $P(s_i)$. Then, we assume that each sample is drawn from the following distribution:

$$\bar{P}(\boldsymbol{X}, \bar{\boldsymbol{Y}}) = \sum_{j=1}^{c-1} \bar{P}(\boldsymbol{X}, \bar{\boldsymbol{Y}} | s_i = j) P(s_i = j), \tag{48}$$

where

$$\bar{P}(\boldsymbol{X}, \bar{\boldsymbol{Y}} | s_i = j) P(s_i = j) := \left\{ \begin{array}{ll} \frac{1}{\binom{c-1}{j}} \sum_{\boldsymbol{Y} \notin \bar{Y}_i} P(\boldsymbol{X}, \boldsymbol{Y}), & \text{if} \quad s_i = j, \\ 0, & \text{otherwise.} \end{array} \right\}. \tag{49}$$

## F    EXPERIMENTAL DETAILS

**Details of Compared Methods.**

- UB-EXP (Feng et al., 2020), an unbiased risk estimator with an estimation error bound, which is derived for Multiple CLL specially.
- UB-LOG (Feng et al., 2020), another unbiased risk estimator with an estimation error bound but with a different multi-class classification loss function.

- SCL-EXP (Chou et al., 2020), a surrogate complementary loss with the use of exponential loss function.

- SCL-LOG (Chou et al., 2020), a surrogate complementary loss with the use of negative log loss function.

- POCR (Wang et al., 2021), an algorithm which combines the SCL-LOG loss and the consistency regularization technique.

- SELF-CL (Liu et al., 2022), a self-supervised learning algorithm which integrates self-distillation to CLL.

- ComCo (Jiang et al., 2024a), a contrastive learning framework which leverages the contrastive learning technique on CLL.

**Datasets generation.** To generate single complementary label, we randomly select one of the complementary classes per instance. To generate multiple complementary labels, let $s$ be the size of $\bar{Y}$, we first instantiate $p(s) = \binom{K-1}{s}/(2^K - 2), s \in \{1, 2, ..., K-1\}$, which represents the possibility of randomly sample a complementary label set whose size is $s$ from all possible complementary label sets which has $2^K - 2$ sets to choose from. Then for each instance $x_i$, we first sample $s_i$ from $p(s_i)$, and then sample a complementary label set $\bar{Y}_i$ with size $s$ from $p(\bar{Y}_i) = 1/\binom{K-1}{s_i}$.

**Implementation.** Values of hyperparameters in PLNL are set as follows. The queue size $t$ is selected from $\{2, 3, 4, 5\}$, $k$-NN parameter $k$ is selected from $\{100, 250, 500\}$. The $\alpha$ in the instance-aware self-adaptive threshold is selected from $\{0.1, 0.3, 0.5, 0.7, 0.9, 0.99\}$. For each method, we train the commonly used PreAct-ResNet18 (He et al., 2016) with 200 epochs (initial 20 epochs for warm-up), and use SGD as the opimizer with a momentum of 0.9, a weight decay of 1e-4. We set the batch size from $\{64, 128\}$, the initial learning rate from $\{10^{-1}, 10^{-2}\}$, and we use cosine learning rate scheduling with final learning rate $10^{-3}$. We employed *faiss* (Johnson et al., 2021) to compute $k$-NN instances in the output space, which is a library for efficient similarity search and clustering of dense vectors. For weak augmentations, we employ normalization, horizontal flipping and random cropping. For strong augmentations, we use RandAugment strategy for all, which selects the type and magnitude of augmentation based on uniform probability.

For implementation of SSL methods, we firstly pre-train the network with complete CLL dataset for 200 epochs. Next, we perform pseudo-labeling iteratively and train the model for another 200 epochs.

All of the experiments are implemented based on PyTorch (Paszke et al., 2019) and all of our experiments are conducted with 8 NVIDIA 4090 GPUs.

# G  MORE EXPERIMENTAL RESULTS

## G.1  PARAMETER SENSITIVITY ANALYSIS

**Influence of memory bank size $t$.** As shown in Table 5, it is obvious that $t$ has little effect on the experimental results on the three datasets. We chose $t = 5$, which has a slightly better effect. It is important to utilize historical confidence information to alleviate confirmation bias, but the size of the memory banks does not matter.

**Influence of confidence threshold $\lambda$.** From Table 6, we observe that there is a trade-off between $\eta$ and $1$-$\epsilon_1$, i.e., a higher threshold will lead the precision to increase but result in less reliable instances selected while a lower threshold will decrease the precision but select more reliable instances. Instance-aware self-adaptive threshold (IST) shows obvious performance gain compared with a fixed global threshold, showcasing its effectiveness.

Table 5: Classification accuracy of PLNL with different memory bank size $t$ on three benchmark datasets. The best results are highlighted in bold and the second best are underlined (The same applies hereinafter).

| t | STL-10 SCLL | CIFAR-10 SCLL | CIFAR-100 MCLL |
|---|---|---|---|
| 2 | 54.45% | 94.12% | 64.21% |
| 3 | 54.95% | 93.98% | 63.95% |
| 4 | 54.88% | 94.56% | 63.81% |
| 5 | **55.25%** | **94.78%** | **64.33%** |

Table 6: Classification accuracy and PLG performance of PLNL with different confidence threshold $\lambda$ on two benchmark datasets. $\eta$ denotes PLG selected ratio, $1 - \epsilon_1$ denotes PLG precision, $1 - \epsilon_2$ denotes NLE precision. IST denotes instance-aware self-adaptive threshold.

| $\lambda$ | CIFAR-10, SCLL | | | CIFAR-100, MCLL | | |
|---|---|---|---|---|---|---|
| | Accuracy | $\eta$ | 1-$\epsilon_1$ | Accuracy | $\eta$ | 1-$\epsilon_1$ |
| 0.85 | 92.89% | 99.86% | 90.79% | 62.12% | 81.52% | 72.88% |
| 0.90 | 93.85% | 95.78% | 92.43% | 63.95% | 77.25% | 77.34% |
| 0.95 | 94.74% | 87.28% | 98.68% | 64.12% | 73.48% | 83.61% |
| IST ($\alpha = 0.5$) | **94.78%** | 93.40% | 96.79% | **64.33%** | 76.08% | 79.84% |

**Influence of $k$-NN parameter $k$.** As shown in Table 7, the selection of $k$ depends on the specific situation of the dataset. For example, STL-10 has only 500 labeled instances for each category. If $k$ is set too large, there will be instances that are not in the category near the decision boundary, which will induce more noise in NLE and cause performance degradation.

Table 7: Classification accuracy of PLNL with different $k$-NN parameter $k$ on three benchmark datasets.

| $k$ | STL-10 SCLL | CIFAR-10 SCLL | CIFAR-100 MCLL |
|---|---|---|---|
| 100 | 54.45% | 94.25% | 63.27% |
| 250 | **55.25%** | 94.73% | **64.33%** |
| 500 | 47.23% | **94.78%** | 59.65% |

## G.2 QUANTATIVE RESULTS: PERFORMANCE OF PLG AND NLE

As shown in Table 8, PLG achieves high selected ratio in all datasets, even reaching 98.86% on FMNIST (MCLL), and 76.08% on the difficult CIFAR-100. And the precision remains high even at a high selected ratio. NLE also has high precision, reaching 80.84% even on CIFAR-100.

Table 8: The performance of PLG and NLE on three settings. The results (mean $\pm$ std) are reported over 3 random trials. $\eta$ denotes PLG selected ratio, $1 - \epsilon_1$ denotes PLG precision, $1 - \epsilon_2$ denotes NLE precision.

| Dataset | Case | Performance | | |
|---|---|---|---|---|
| | | $\eta$ | 1-$\epsilon_1$ | 1-$\epsilon_2$ |
| CIFAR-10 | SCLL | 93.40±0.14% | 96.79±0.08% | 97.79±0.09% |
| | MCLL | 97.49±0.12% | 99.30±0.05% | 98.91±0.07% |
| FMNIST | SCLL | 97.25±0.08% | 95.24±0.11% | 97.56±0.08% |
| | MCLL | 98.86±0.06% | 97.89±0.11% | 98.97±0.03% |
| CIFAR-100 | MCLL | 76.08±0.22% | 79.84±0.41% | 80.84±0.04% |

## G.3 EXPERIMENTAL RESULTS ON TINY-IMAGENET

Tiny-ImageNet (Le and Yang, 2015) contains 100000 images of 200 classes. Each class has 500 training images, 50 validation images and 50 test images. Due to its huge number of categories, it is an extremely difficult dataset for complementary label learning. Most existing CLL methods have only tested their performance on 10-class small datasets. Most of their backbones are ResNet and a single complementary label is a rather difficult setting for large datasets, these will lead to poor performance of traditional CLL methods.

However, we have tested the performance of our method on Tiny-ImageNet with MCLL settings. The results are shown in Table 9. Though most of the methods perform poorly, our method still outperforms traditional CLL methods obtrusively.

Table 9: Comparison of classification accuracies between different methods on Tiny-ImageNet with multiple complementary labels per instance.

| Method | Tiny-ImageNet |
|--------|---------------|
| UB-EXP | 3.89% |
| UB-LOG | 7.17% |
| SCL-EXP | 3.36% |
| SCL-LOG | 8.96% |
| POCR | 4.29% |
| SELF-CL | 7.87% |
| ComCo | 8.52% |
| Ours | **11.87%** |

### G.4 EXPERIMENTAL RESULTS ON REAL-WORLD DATASETS (CLCIFAR-10 AND CLCIFAR-20).

For real-world scenarios, we added two datasets, including CLCIFAR-10 and CLCIFAR-20 (Wang et al., 2023a), which are two manually-annotated datasets specially designed for CLL. The results are shown in Table 10.

Table 10: Comparison of classification accuracies between different methods on CLCIFAR-10 and CLCIFAR-20.

| Method | CLCIFAR-10 | CLCIFAR-20 |
|--------|------------|------------|
| UB-EXP | 41.95% | 8.08% |
| UB-LOG | 38.46% | 6.00% |
| SCL-EXP | 40.93% | 7.86% |
| SCL-LOG | 38.77% | 7.45% |
| POCR | 51.83% | 8.82% |
| SELF-CL | 44.35% | 8.64% |
| ComCo | 46.42% | 8.21% |
| Ours | **64.39%** | **9.88%** |

It can be seen from the above table that PLNL performs much better than previous CLL methods in real-world scenarios, with a $12.56\%$ improvement on CLCIFAR-10 and a $1.06\%$ improvement on CLCIFAR-20 compared to previous SOTA (POCR).

### G.5 EXPERIMENTAL RESULTS UNDER BIASED COMPLEMENTARY LABEL LEARNING SETTINGS

We add the experimental results of three datasets (STL-10, CIFAR-10, CIFAR-100) under biased complementary label learning settings.

To generate biased complementary labels, we first give the probability of each complementary label to be selected, following the same settings in (Yu et al., 2018), where two settings are given, including "without 0" and "with 0".

In the "without 0" setting, for each class $y$, we first randomly split $Y \setminus y$ into three subsets, each containing three elements. Then, for each complementary label in these three subsets, the probabilities are set to $\frac{0.6}{3}$, $\frac{0.3}{3}$, and $\frac{0.1}{3}$, respectively.

In the "with 0" setting, for each class $y$, we first randomly selected three labels in $Y \setminus y$, and then randomly assign them with three probabilities whose summation is 1.

The experimental results are shown in Table 11.

The results reveal two main points:

Table 11: Comparison of classification accuracies between different methods under biased complementary label learning settings.

| Method | STL-10 | CIFAR-10 | CIFAR-100 |
|---|---|---|---|
| **Without 0:** | | | |
| UB-EXP | 52.43% | 22.54% | 1.18% |
| UB-LOG | 48.36% | 22.71% | 1.00% |
| SCL-EXP | 60.85% | 23.53% | 1.07% |
| SCL-LOG | 58.63% | 22.96% | 0.93% |
| POCR | 74.78% | 29.03% | 1.00% |
| PLNL | **75.38%** | **38.82%** | 1.00% |
| **With 0:** | | | |
| UB-EXP | 60.71% | 78.73% | 1.00% |
| UB-LOG | 44.42% | 78.69% | 1.00% |
| SCL-EXP | 61.24% | 79.17% | 1.00% |
| SCL-LOG | 63.36% | 81.51% | 1.00% |
| POCR | 74.69% | 94.71% | 1.00% |
| PLNL | **74.88%** | **95.15%** | 1.00% |

1. The advantage of PLNL over previous CLL methods still exists in biased settings, which demonstrates the robustness of PLNL. For example, under CIFAR-10 ("without 0"), PLNL outperforms previous SOTA by $9.79\%$.

2. All CLL methods cannot converge on biased CIFAR-100.

## G.6 ABLATION STUDY ON THE DIFFERENT NLE STRATEGY ON MODERATELY-CONFIDENT AND UNDER-CONFIDENT

In our method, dividing the dataset into three parts is much more natural than two parts. Since we train a two-view network for instance selection, it naturally divides the whole dataset into three parts based on the number of the output confidences of the two networks meeting the selection criteria:

- Highly-confident: both of two confidences meet the criteria

- Moderately-confident: one of the two confidences meets the criteria

- Under-confident: both two confidences do not meet the criteria

Since the model is under-learned for under-confident instances, the feature space of under-confident instances might be inaccurate, performing NLE based on this would lead to considerable errors. We show the NLE error rate of two strategies in the following table. "PLNL w/ same strategy" denotes treating the moderately-confident and under-confident instances the same way.

Table 12: Comparison of NLE error rate $\epsilon_2$ between different strategies of treating the moderately-confident and under-confident instances.

| Method | CIFAR-10 (SCLL) | CIFAR-10 (MCLL) | CIFAR-100 (MCLL) |
|---|---|---|---|
| PLNL w/ same strategy | $3.42 \pm 0.06\%$ | $2.11 \pm 0.12\%$ | $25.36 \pm 0.09\%$ |
| PLNL | $\mathbf{2.21 \pm 0.09\%}$ | $\mathbf{1.09 \pm 0.07\%}$ | $\mathbf{19.16 \pm 0.04\%}$ |

And the final classification accuracy increases significantly under PLNL strategy.

It can be seen from the above tables that the separate treatment strategy used by PLNL outperforms the same treatment strategy.

Table 13: Comparison of classification accuracies between different strategies of treating the moderately-confident and under-confident instances.

| Method | CIFAR-10 (SCLL) | CIFAR-10 (MCLL) | CIFAR-100 (MCLL) |
|---|---|---|---|
| PLNL w/ same strategy | $93.98 \pm 0.15\%$ | $95.89 \pm 0.24\%$ | $62.25 \pm 0.34\%$ |
| PLNL | $\mathbf{94.78 \pm 0.12\%}$ | $\mathbf{96.80 \pm 0.28\%}$ | $\mathbf{64.33 \pm 0.43\%}$ |

### G.7 EXTRACT FEATURES BASED ON VISUAL LANGUAGE MODELS (VLMS) AND SELF-SUPERVISED LEARNING (SSL) TECHNIQUES

In the paper, we compute $k$-NN instances based on the model output space information, namely the feature extracted by the model itself. It strikes us that different features may have much to do with the NLE performance. To confirm this, we employ different feature extractors for computing $k$-NN, including PreActResNet-18-MoCo, BLIP-2 (Li et al., 2023). Note that these results are just for performance comparison which has nothing to do with the results presented in the main body of the paper. For MoCo, we train a PreActResNet by self-supervised learning method MoCo (He et al., 2020) without any supervision. The weak and strong data augmentations used in MoCo follow the original configurations mentioned in the main body. Then we compute $k$-NN on the 512-dimensional feature output of the PreActResNet. For BLIP-2, we first employ the visual encoder to extract 768-dimensional high-quality representations and then leverage *faiss* (Johnson et al., 2021) to compute $k$-NN instances in this feature space. We compute the average precision of NLE $1 - \epsilon_2$ and accuracy on CIFAR-10 (SCLL) and CIFAR-100 (MCLL). As shown in Table 14, the feature extracted from BLIP-2 outperforms MoCo and ResNet itself significantly. This shows the powerful visual representation ability of VLMs, which has a great potential for facilitating innovation in weakly-supervised learning in the future.

Table 14: Comparison of classification accuracies and NLE precision between different methods on CIFAR-100 with multiple complementary labels per instance. PreActResNet-18 denotes leveraging the model output space information for $k$-NN calculation

| Feature Extractor | CIFAR-10, SCLL | | CIFAR-100, MCLL | |
|---|---|---|---|---|
| | Accuracy | $1 - \epsilon_2$ | Accuracy | $1 - \epsilon_2$ |
| MoCo | 93.12% | 95.64% | 61.82% | 75.34% |
| BLIP-2 | **95.84%** | **99.91%** | **69.85%** | **93.34%** |
| PreActResNet-18 | 94.78% | 97.79% | 64.33% | 80.84% |

### G.8 COMPARISON WITH SEMIREWARD

We further compare the performance of PLNL with one of the recently published SSL method SemiReward (Li et al., 2024). It can be seen that PLNL still outperforms SemiReward significantly in selection ratio and average size of NLS.

## H DEVIATION REPORTS OF TABLE 3 AND TABLE 4

We re-tested and added deviation terms in Table 3 and Table 4 of the paper, with results respectively shown in Table 15 and Table 16.

Table 15: The performance of PLNL with single network on two settings.

| Method | CIFAR-10 SCLL | | CIFAR-100 MCLL | |
|---|---|---|---|---|
| | $\eta$ | $1-\epsilon_1$ | $\eta$ | $1-\epsilon_1$ |
| Single | 84.65±0.12% | 90.21±0.09% | 67.43±0.24% | 70.62±0.31% |
| Two-view | **93.40±0.14%** | **96.79±0.08%** | **76.08±0.22%** | **79.84±0.41%** |

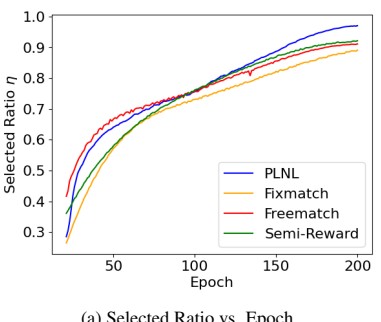
(a) Selected Ratio vs. Epoch

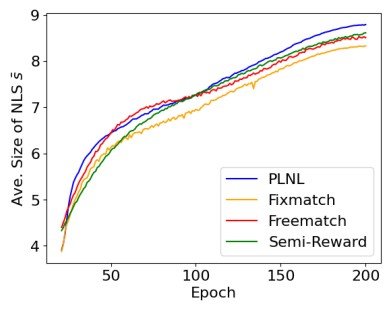
(b) Ave. Num. of NLS vs. Epoch

Figure 4: The experiments is conducted on CIFAR-10 with single complementary labels (SCLL). (a) shows that selected ratio of PLNL trenscends Fixmatch, Freematch and SemiReward significantly. (b) shows that average size of NLS of PLNL is significantly larger due to specially designed technique NLE for enhancing the untrustworthy negative labels. Nearly all negative labels are revealed at the end of training, almost reaching 9 negative labels for each instances in CIFAR-10.

Table 16: Classification accuracy of degenerated methods on three settings.

| Method | STL-10 SCLL | CIFAR-10 SCLL | CIFAR-100 MCLL |
|---|---|---|---|
| PLNL | **55.25±0.35%** | **94.78±0.12%** | **64.33±0.43%** |
| PLNL v1 | 49.25±0.41% | 93.75±0.08% | 63.09±0.27% |
| PLNL v2 | 49.82±0.39% | 92.01±0.12% | 58.94±0.31% |
| PLNL v3 | 53.22±0.35% | 94.28±0.10% | 63.14±0.32% |
| POCR | 34.96±0.32% | 94.15±0.09% | 53.16±0.11% |

# I    DETAILED DERIVATION OF EQ. (30)

$$\widehat{R}'(f) = \frac{1}{N}\sum_{i=1}^{N}\bar{\mathcal{L}}_{CLL}(f(\boldsymbol{x}_i), \widehat{\widehat{Y}}_i),$$

$$= \frac{1}{N}\sum_{i=1}^{N}\sum_{y\notin\widehat{Y}_i}\frac{1}{K-|\widehat{\widehat{Y}}_i|}\ell(f(\boldsymbol{x}), y). \tag{50}$$

This step is due to the definition of CLL loss function:

$$\mathcal{L}_{CLL}(f(\boldsymbol{x}), \widehat{\widehat{Y}}_i) = \sum_{y\notin\widehat{Y}_i}\frac{1}{K-|\widehat{\widehat{Y}}_i|}\ell(f(\boldsymbol{x}), y).$$

Then we have:

$$\widehat{R}'(f) = \frac{1}{N}\sum_{i=1}^{N}\ell(f(\boldsymbol{x}_i), y_i) + \frac{1}{N}\sum_{i=1}^{N}\mathbb{I}(y_i\notin\widehat{\widehat{Y}}_i)\left[\sum_{c\notin\widehat{Y}_i, c\neq y_i}\frac{1}{K-|\widehat{\widehat{Y}}_i|}\ell(f(\boldsymbol{x}_i), c) - \underline{\frac{K-|\widehat{\widehat{Y}}_i|-1}{K-|\widehat{\widehat{Y}}_i|}\ell(f(\boldsymbol{x}_i), y_i)}\right]$$

$$+ \frac{1}{N}\sum_{i=1}^{N}\mathbb{I}(y_i\in\widehat{\widehat{Y}}_i)\left[\sum_{c\notin\widehat{Y}_i}\frac{1}{K-|\widehat{\widehat{Y}}_i|}\ell(f(\boldsymbol{x}_i), c) \underline{\underline{-\ell(f(\boldsymbol{x}_i), y_i)}}\right]. \tag{51}$$

The first term is the loss on the ground-truth label, summing up to the empirical risk $\widehat{R}(f)$. The second and third term is the difference between empirical risk $\widehat{R}(f)$ and practical empirical risk with pseudo-labeling error $\widehat{R}'(f)$.

Since pseudo-labeling error may occur during PLG and NLE processes, the ground-truth label of $\boldsymbol{x}_i$ may be mistakenly included or correctly excluded in the enhanced negative label set $\widehat{\overline{Y}}_i$. We discuss two situations separately (i.e. $\mathbb{I}(y_i \in \widehat{\overline{Y}}_i)$ and $\mathbb{I}(y_i \notin \widehat{\overline{Y}}_i)$). We extract the empirical risk term $\widehat{R}(f)$ by subtracting the equivalent value in the second and third terms, which is shown between square brackets above.

As $\ell(f(\boldsymbol{x}), y)$ is bounded by a positive value $B$. We can scale up the second and third terms by directly removing the terms after minus sign, which is double underlined above, which will yield the following line.

$$\widehat{R}'(f) \leq \widehat{R}(f) + \frac{1}{N} \sum_{i=1}^{N} \mathbb{I}(y_i \notin \widehat{\overline{Y}}_i) \sum_{c \notin \widehat{\overline{Y}}_i, c \neq y_i} \frac{1}{K - |\widehat{\overline{Y}}_i|} \ell(f(\boldsymbol{x}_i), c),$$
$$+ \frac{1}{N} \sum_{i=1}^{N} \mathbb{I}(y_i \in \widehat{\overline{Y}}_i) \sum_{c \notin \widehat{\overline{Y}}_i} \frac{1}{K - |\widehat{\overline{Y}}_i|} \ell(f(\boldsymbol{x}_i), c). \tag{52}$$

Then we utilize Jensen's Inequality for further scaling up.

Jensen's Inequality for concave function $\varphi$:

$$\varphi\left(\sum_{i=1}^{n} g(x_i)\lambda_i\right) \geq \sum_{i=1}^{n} \varphi(g(x_i))\lambda_i,$$

where $\lambda_1 + \lambda_2 + \cdots + \lambda_n = 1, \lambda_i \geq 0$.

We first scale the second term in Eq. (52). Here, the concave function is $\varphi(|\widehat{\overline{Y}}_i|) = \frac{1}{K - |\widehat{\overline{Y}}_i|}$, where $\lambda_i = \frac{1}{N}$, since $\sum_{c \notin \widehat{\overline{Y}}_i, c \neq y_i} \ell(f(\boldsymbol{x}_i), c)$ indicates the sum of binary losses on non-complementary labels $c \notin \widehat{\overline{Y}}_i$, excluding the ground-truth label $c \neq y_i$. The number of non-complementary labels is $K - |\widehat{\overline{Y}}_i|$, excluding the ground-truth label will get $K - |\widehat{\overline{Y}}_i| - 1$, which means computing the binary loss $\ell(f(\boldsymbol{x}_i), c)$ for $K - |\widehat{\overline{Y}}_i| - 1$ times. Finally, since $\ell$ is bounded by $B$, we can make such a scale:

$$\frac{1}{N} \sum_{i=1}^{N} \sum_{c \notin \widehat{\overline{Y}}_i, c \neq y_i} \frac{1}{K - |\widehat{\overline{Y}}_i|} \ell(f(\boldsymbol{x}_i), c)$$
$$\leq \frac{1}{N} \sum_{i=1}^{N} \frac{K - |\widehat{\overline{Y}}_i| - 1}{K - |\widehat{\overline{Y}}_i|} B,$$
$$\leq \frac{K - \frac{1}{N} \sum_{i=1}^{N} |\widehat{\overline{Y}}_i| - 1}{K - \frac{1}{N} \sum_{i=1}^{N} |\widehat{\overline{Y}}_i|} B. \tag{53}$$

Since we have defined that $\bar{s} = \frac{1}{N} \sum_{i=1}^{N} |\widehat{\overline{Y}}_i|$ and $\mathbb{I}(y_i \notin \widehat{\overline{Y}}_i) = 1 - \epsilon$, we have

$$\frac{1}{N} \sum_{i=1}^{N} \mathbb{I}(y_i \notin \widehat{\overline{Y}}_i) \frac{K - \frac{1}{N} \sum_{i=1}^{N} |\widehat{\overline{Y}}_i| - 1}{K - \frac{1}{N} \sum_{i=1}^{N} |\widehat{\overline{Y}}_i|} B$$
$$\leq (1 - \epsilon) \frac{K - \bar{s} - 1}{K - \bar{s}} B. \tag{54}$$

The third term in Eq. (52) follows similar scaling procedures.

## J    COMPARISON BETWEEN CLL AND PARTIAL LABEL LEARNING (PLL)

**CLL relieves the efforts for Data Annotation.**    CLL and partial label learning (PLL) distincts intrinsically in the data annotation stage. From the perspective of data annotation, PLL still requires true label being selected while CLL only requires classes the instance does not belong to, which makes data annotation less laborious.

Moreover, CLL data annotation reduce the complex process of labeling data into a much easier "yes/no" question. For each instance, answering a "yes/no" question naturally gives either a true label or a complementary label. This logic of creating datasets is creative and time-efficient, which has great potential in the future.

Most importantly, CLL certainly provides machine learning community a new perspective of viewing data annotation and label distribution.

**CLL shows promise in Privacy Field.**    The formulation of learning from complementary labels may also be useful in the context of *privacy-aware machine learning*: a subject needs to answer private questions such as psychological counseling which can make him/her hesitate to answer directly. In such a situation, providing a complementary label, i.e., one of the incorrect answers to the question, would be *mentally less demanding*. The issue will surely be investigated in the future.

**CLL and PLL are indeed related.**    CLL and PLL can enlighten and learn from each other in terms of methodology, etc.

In conclusion, the problem settings are intrinsically different. Both CLL and PLL is reasonable and deserves further investigation and research.

**The advantages of PLNL over existing PLL methods (motivationally and experimentally).** Experimentally, we supplement two SOTA partial label learning (PLL) methods: DPLL (Wu et al., 2022) and PiCO+ (Wang et al., 2024) for comparison. The results are shown in Table 17 as follows.

**Note that for the transformation from CL datasets to PL datasets, we conduct this by simply taking the complement of complementary label sets as candidate label sets.**

Table 17: Results comparing DPLL, PiCO+, and PLNL across multiple test cases.

| Method | CIFAR-10 SCLL | CIFAR-10 MCLL | CIFAR-100 SCLL | CIFAR-100 MCLL |
|---|---|---|---|---|
| DPLL | 93.98±0.09% | 95.49±0.11% | 1.00% | 61.47±0.21% |
| PiCO+ | 94.43±0.07% | 95.35±0.08% | 1.00% | 61.25±0.32% |
| PLNL | **94.78±0.12%** | **96.80±0.28%** | 1.00% | **64.33±0.43%** |

It can be seen from the results that PLNL outperforms PLL SOTAs in all test cases.

