# OpenReview forum: "Complementary Label Learning with Positive Label Guessing and Negative Label Enhancement"
_ICLR.cc/2025/Conference — ICLR 2025 Poster_

### Official Review · Reviewer_zFpU · 2024-10-28

**Soundness:** 3
**Presentation:** 3
**Contribution:** 2
**Rating:** 5
**Confidence:** 4

**Summary:**

This paper proposes to decompose the CLL problem into multi-class classification problems. The one is to infer the positive label by a positive label guessing method, and another is to infer the negative labels by enhancing the negative labels of the moderately-confident and under-confident instances. Besides, this paper provides the upper bounds for the error rate of positive label guessing method and negative label enhancement method, and the effectiveness of the proposal is demonstrated by several experiments.

**Strengths:**

This paper proposes a new method for CLL, which divides the CLL into two subtasks: PLG and NLE to deal with the high-confident and low-confident instances separately. The authors also verify the effectiveness of the proposal by several experiments and theorems.

**Weaknesses:**

1. This paper attempts to learn a multi-label classifier from CL. However, the difference between CL and partial labels is not clarified. Initially, Ishida proposed complementary label learning, and noted that complementary label y can be regarded as an extreme case of partial labels given to all K −1 classes other than class y. However, in lines 48 and 121 of this paper, the authors claim that CL can be multiple classes that the instance does not belong to, which seems to fundamentally eliminate the difference between partial label and CL.
2. This paper does not well-motivate the proposed method, so that its advantages over existing methods are not clear. It is also not clear what conditions make the proposed method work or fail. Besides, given the strong relevance of this paper and partial label learning, it is not yet clear what advantages the algorithms in this paper have over existing partial label learning algorithms, either motivationally or experimentally.
3. Theorem 2 and Theorem 3 do not give valuable information. For example, the theorem presented in Equation (21) seems to say one thing that a probability value does not exceed 1. Furthermore, the author's discussion of Theorems 2 and 3 is very shallow; they just tell readers that there is an upper bound on the error rate, which is meaningless because any error rate has an upper bound of 1.

**Questions:**

1. What is the difference between partial label learning and the problem addressed in this paper?
2. What are the advantages of the proposed algorithm over other existing algorithms (including CLL and partial label learning)?
3. What are the conditions for the proposed algorithm?
4. What do Theorem 2 and Theorem 3 reveal?

---

> ### Author Response · Authors · 2024-11-22
> **Response to Reviewer zFpU PART 1**
>
> Thank you so much for your insightful comments!
>
> **W1: This paper attempts to learn a multi-label classifier from CL. However, the difference between CL and partial labels is not clarified. Initially, Ishida proposed complementary label learning, and noted that complementary label y can be regarded as an extreme case of partial labels given to all K −1 classes other than class y. However, in lines 48 and 121 of this paper, the authors claim that CL can be multiple classes that the instance does not belong to, which seems to fundamentally eliminate the difference between partial label and CL.**
>
> A: Thank you for your detailed questions! We will answer you questions in the following three aspects.
>
> **a. CLL relieves the efforts for Data Annotation**
>
> CLL and partial label learning (PLL) distincts intrinsically in the data annotation stage. From the perspective of data annotation, PLL still requires true label being selected while CLL only requires classes the instance does not belong to, which makes data annotation less laborious.
>
> Moreover, CLL data annotation reduce the complex process of labeling data into a much easier "yes/no" question. For each instance, answering a "yes/no" question naturally gives either a true label or a complementary label. This logic of creating datasets is creative and time-efficient, which has great potential in the future.
>
> Most Importantly, CLL certainly provides machine learning community a new perspective of viewing data annotation and label distribution.
>
> **b. CLL shows promise in Privacy Field**
>
> The formulation of learning from complementary labels may also be useful in the context of **privacy-aware machine learning**: a subject needs to answer private questions such as psychological counseling which can make him/her hesitate to answer directly. In such a situation, providing a complementary label, i.e., one of the incorrect answers to the question, would be **mentally less demanding**. The issue will surely be investigated in the future.
>
> **c. CLL and PLL are indeed related**
>
> CLL and PLL can enlight and learn from each other in terms of methodology, etc.
>
> In conclusion, the problem settings are intrinsically different. Both CLL and PLL is reasonable and deserves further investigation and research.
>
> **W2: This paper does not well-motivate the proposed method, so that its advantages over existing methods are not clear. It is also not clear what conditions make the proposed method work or fail. Besides, given the strong relevance of this paper and partial label learning, it is not yet clear what advantages the algorithms in this paper have over existing partial label learning algorithms, either motivationally or experimentally.**
>
> A: Thank you for your detailed questions! We will answer your questions with the following three aspects.
>
> **a. The Motivation of PLNL**
>
> Previous studies on CLL can be roughly divided into two categories: methods that attempt to construct
> an unbiased risk estimator (URE-based) (Ishida et al., 2017; 2019; Feng et al., 2020) and methods
> based on feature learning (FL-based) (Chou et al., 2020; Wang et al., 2021; Liu et al., 2022; Jiang
> et al., 2024). These methods either focus on the design of robust loss functions or the exploration of representation learning, while neglecting the power of **output space information**, namely **confidence information**.
>
> Previous studies in semi-supervied learning (SSL) have already paved the way for confidence-based sample selection methods. However, previous SSL methods lack **the utilization of untrustworthy instances**. They either discard this part or simply employ techniques such as consistency regularization.
>
> We make the first attempt to **leverage the output space information (PLG)** to facilitate CLL, combining with **well-designed utilization of untrustworthy instances (NLE)**, to make up the whole framework of PLNL. Last but not least, we unify the pseudo-labeling methods in a negative label recovery framework and provide **theoretical analysis of PLNL**.

---

> ### Author Response · Authors · 2024-11-22
> **Response to Reviewer zFpU PART 2**
>
> **b. The conditions where PLNL works or fails**
>
> First, abundant experimental results show that PLNL is neither a parameter-sensitive method  nor a method that requires strict conditions.
>
> Second, in Theorem 1, we theoretically prove that the generalization error of PLNL is bounded.
>
> Suppose that $\ell(f(\boldsymbol{x}), y)$ is bounded by $B$. For pseudo-labeling error rate $\epsilon \in (0,1)$, for any $\delta >0$, with probability at least $1-\delta$, we have
>
> $$
>         R(\hat{f})-R(f^{*})\le 2(1-\frac{1-\epsilon}{K-\bar{s}})B+4\rho K\mathfrak{R}_{N}(\mathcal{F})+2KB\sqrt{\frac{log\frac{2}{\delta}}{2N}},
> $$
>
> From Theorem 1, we can see that in order that the error bound reaches zero, the most important factor is the PLNL error rate $\epsilon$, which consists of PLG error rate $\epsilon_1$ and NLE error rate $\epsilon_2$.
>
> Furthermore, we give Theorem 2 and Theorem 3 for further bounding the PLG error rate $\epsilon_1$ and NLE error rate $\epsilon_2$. We theoretically bound them to small numbers close to zero under mild assumptions. **Whether these assumptions are met influences whether PLNL works or fails.** Concretely, controlling PLG error rate $\epsilon_1$ and NLE error rate $\epsilon_2$ will help PLNL work better.
>
> Empirically, PLNL **does not require detailed tunning of hyper-parameters** and it is **robust under biased settings**. Firstly, our ablation study in **Appendix G.2.** has proved that hyper-parameters have little to do with the classification performance of PLNL and hence no detailed manual tuning of hyper-parameters is required. Secondly, we add the experimental results of three datasets (STL-10, CIFAR-10, CIFAR-100) under biased complementary label learning settings.
>
> To generate biased complementary labels, we first give the probability of each complementary label to be selected, following the same settings in [1], where two settings are given, including "without 0" and "with 0".
>
> In the “without0” setting, for each class $y$, we first randomly split $Y \textbackslash\{y\}$ to three subsets, each containing three elements. Then, for each complementary label in these three subsets, the probabilities are set to $\frac{0.6}{3}$ , $\frac{0.3}{3}$ , and $\frac{0.1}{3}$,  respectively.
>
> In the “with0” setting, for each class $y$, we first randomly selected three labels in $Y \textbackslash\{y\}$, and then randomly assign them with three probabilities whose summation is 1.
>
> **Without 0:**
>
> | Method  | STL-10     | CIFAR-10   | CIFAR-100 |
> | ------- | ---------- | ---------- | --------- |
> | UB-EXP  | 52.43%     | 22.54%     | **1.18%** |
> | UB-LOG  | 48.36%     | 22.71%     | 1.00%     |
> | SCL-EXP | 60.85%     | 23.53%     | 1.07%     |
> | SCL-LOG | 58.63%     | 22.96%     | 0.93%     |
> | POCR    | 74.78%     | 29.03%     | 1.00%     |
> | PLNL    | **75.38%** | **38.82%** | 1.00%     |
>
> **With 0:**
>
> | Method  | STL-10     | CIFAR-10   | CIFAR-100 |
> | ------- | ---------- | ---------- | --------- |
> | UB-EXP  | 60.71%     | 78.73%     | 1.00%     |
> | UB-LOG  | 44.42%     | 78.69%     | 1.00%     |
> | SCL-EXP | 61.24%     | 79.17%     | 1.00%     |
> | SCL-LOG | 63.36%     | 81.51%     | 1.00%     |
> | POCR    | 74.69%     | 94.71%     | 1.00%     |
> | PLNL    | **74.88%** | **95.15%** | 1.00%     |
>
> The results reveal two main points:
>
> 1. The advantage of PLNL over previous CLL methods still exists in biased settings, which demonstrates the robustness of PLNL. For example, under CIFAR-10 "without 0", PLNL outperforms previous SOTA by $9.79\%$.
> 2. All CLL methods cannot converge on biased CIFAR-100.
>
> We also show the performance degradation of different CLL methods under biased settings compared to uniform settings. The results actually show that PLNL is robust than previous SOTA CLL method POCR.
>
> | Method  | CIFAR-10<br />Uniform | CIFAR-10<br />Biased ('without 0') | Difference |
> | ------- | --------------------- | ---------------------------------- | ---------- |
> | UB-EXP  | 62.90%                | 22.54%                             | 40.36%     |
> | UB-LOG  | 70.28%                | 22.71%                             | 47.57%     |
> | SCL-EXP | 72.35%                | 23.53%                             | 48.82%     |
> | SCL-LOG | 79.87%                | 22.96%                             | 56.91%     |
> | POCR    | 94.15%                | 29.03%                             | 65.12%     |
> | PLNL    | **94.78%**            | **38.82%**                         | 55.96%     |

---

> ### Author Response · Authors · 2024-11-22
> **Response to Reviewer zFpU PART 3**
>
> **c. The advantages of PLNL over existing PLL methods (motivationally and experimentally)**
>
> The differences between PLNL and existing CLL/SSL methods are clearly emphasized in **W2 a.** for your reference.
>
> Experimentally, we supplyment two SOTA partial label learning (PLL) methods: DPLL (ICML 2023) [2] and PiCO+(TPAMI 2024) [3] for comparison.(The results (mean ± std) are reported over 3 random trials.)
>
> For the transformation from CL datasets to PL datasets, we conduct this by simply taking the complement of complementary label sets as candidate label sets.
>
>
> | Method | CIFAR-10<br />SCLL  | CIFAR-10<br />MCLL  | CIFAR-100<br />SCLL | CIFAR-100<br />MCLL |
> | ------ | ------------------- | ------------------- | ------------------- | ------------------- |
> | DPLL   | 93.98$\pm$0.09%     | 95.49$\pm$0.11%     | 1.00%               | 61.47$\pm$0.21%     |
> | PiCO+  | 94.43%$\pm$0.07%    | 95.35$\pm$0.08%     | 1.00%               | 61.25$\pm$0.32%     |
> | PLNL   | **94.78$\pm$0.12%** | **96.80$\pm$0.28%** | 1.00%               | **64.33$\pm$0.43%** |
>
> It can be seen from the results that PLNL outperforms PLL SOTAs in all test cases.
>
> **W3: Theorem 2 and Theorem 3 do not give valuable information. For example, the theorem presented in Equation (21) seems to say one thing that a probability value does not exceed 1. Furthermore, the author's discussion of Theorems 2 and 3 is very shallow; they just tell readers that there is an upper bound on the error rate, which is meaningless because any error rate has an upper bound of 1.**
>
> A: Thank you for your detailed questions on our theoretical analysis! We will show that the error rates are meaningfully bounded by small numbers close to zero both empirically and theoretically. We will also show that the theoretical framework is complete only when Theorem 2 and 3 are given.
>
> **a. Empirical results show that PLG/NLE error rate are low**
>
> Empirically, in fact, we have already showed that both the error rates of PLG and NLE are close to zero. Please refer to Table 5 in Appendix for further information. For example, for CIFAR-10 (SCLL), we showed that the the precision of PLG and NLE reach 96.79$\pm$0.08% and 97.79$\pm$0.09% respectively, which in turn gives very low error rates of approximately 3.21% and 2.21%.
>
> **b. Theoretical results show that the upper bounds for these error rates are small numbers much close to zero.**
>
> **Theorem3:** Let's first take a look at the right side of Eq.(22).
> $$
> \sum_{j=1}^k \binom{|Y_i| - 1}{|Y_i| - \tau_i} F_{\beta_k}(k - j + 1, j)^{(|Y_i| - \tau_i)} b_{\alpha_k}(k, j)
> $$
>
> where:
>
> - $\binom{|Y_i| - 1}{|Y_i| - \tau_i}$ is a bin binomial coefficient, which is a non-negative integer, and in this context, it scales the term.
>
> - $F_{\beta_k}(k(k - j + 1, j)$ $)$ is the regularized incomplete beta function, which lies in the range $[0, 1]$.
>
> - $b_{\alpha_k}(k, j)$ represents the probability mass function of a binomial distribution $B(k, \alpha_k)$, also within $[0, 1]$.
>
>
> Since $F_{\beta_k}(k - j + 1, j)$ is within $[0, 1]$, raising it to a high power, specifically $ (|Y_i| - \tau_i) $, which is assumed to be large in practical scenarios, causes this term to approach zero, making the contribution of each term in the sum negligible.
>
> Thus, the right-hand side of the inequality is approximately:
> $$
> \sum_{j=1}^k \binom{|Y_i| - 1}{|Y_i| - \tau_i} F_{\beta_k}(k - j + 1, j)^{(|Y_i| - \tau_i)} b_{\alpha_k}(k, j) \approx 0
> $$
>
> **Theorem2:** Though the theoretical approximation of $\epsilon_1$ is hard, let's look at Eq.(21) and its conditions.
> $$
> \epsilon_1=\mathbb{P}(y_i \in \widehat{\bar{Y}}_i)\le (K-1-s_i)\psi,
> $$
>
> where $K$ is class number, $s_i$ is the size of $\bar{Y}_i$ and $\psi\in (0,\frac{1}{K-1-s_i})$.
>
> Note that $(0,\frac{1}{K-1-s_i})$ is an open interval so that the worst case, where $\epsilon_1=1$ will always be avoided. From this point, Theorem 3 indicates that performing PLE is not a bad choice.
>
> **c. Theorem 2 and Theorem 3 plays a significant role in the whole theoretical framework.**
>
> In Theorem 1, we theoretically prove that the generalization error of PLNL is bounded.
>
> From Theorem 1, we can see that in order that the error bound reaches zero, the most important factor is the pseudo-labeling error rate $\epsilon$, which consists of PLG error rate $\epsilon_1$ and NLE error rate $\epsilon_2$.
>
> Concretely, controlling PLG error rate $\epsilon_1$ and NLE error rate $\epsilon_2$ will help PLNL work better.
>
> Finally, we theoretically bound them to small numbers close to zero under mild assumptions.

---

> > ### Author Response · Authors · 2024-11-22
> > **Response to Reviewer zFpU PART 4**
> >
> > **Q1: What is the difference between partial label learning and the problem addressed in this paper?**
> >
> > A: Please refer to W1 a.b.c..
> >
> > **Q2: What are the advantages of the proposed algorithm over other existing algorithms (including CLL and partial label learning)?**
> >
> > A:  Please refer to W2 c.
> >
> > **Q3: What are the conditions for the proposed algorithm?**
> >
> > A: Please refer to W2 b.
> >
> > **Q4: What do Theorem 2 and Theorem 3 reveal?**
> >
> > A: Please refer to W3 a.b.c..
> >
> > **Reference**
> >
> > [1] Yu X, Liu T, Gong M, et al. Learning with biased complementary labels[C]//Proceedings of the European conference on computer vision (ECCV). 2018: 68-83.
> >
> > [2] Wu D D, Wang D B, Zhang M L. Revisiting consistency regularization for deep partial label learning[C]//International conference on machine learning. PMLR, 2022: 24212-24225.
> >
> > [3] Wang H, Xiao R, Li Y, et al. Pico+: Contrastive label disambiguation for robust partial label learning[J]. IEEE Transactions on Pattern Analysis and Machine Intelligence, 2023.

---

> ### Author Response · Authors · 2024-11-25
>
> Dear reviewer zFpU,
>
> Thanks again for your time and efforts in reviewing this paper and the valuable comments on improving its quality. As the reviewer-author discussion deadline approaches, please take a few minutes to read the rebuttal. If you have further concerns, we are happy to provide more explanations. Thanks.
>
> Regards from the authors.

---

> > ### Comment · Reviewer_zFpU · 2024-11-27
> >
> > Thanks for the response. Although the authors highlight the differences of generation process between PLL and the CL studied in this paper, they are ultimately the same in terms of label representation and the semantics of each label value. Accordingly, although the authors add experiments to verify the performance advantages of this paper's method over existing PLL methods, it is not clear what shortcomings of existing PLL methods can be addressed by this paper's method.

---

> > > ### Author Response · Authors · 2024-11-27
> > >
> > > Thanks for your thoughtful response. Below, we provide a detailed response to address your concerns:
> > >
> > > **a. Acknowledgement of the Relationship Between CLL and PLL**
> > >
> > > We agree with your perspective that CLL and PLL can be viewed as dual problems. In essence, the complementary labels in CLL and the candidate labels in PLL are complementary sets. This duality indeed enables a conversion between the two problems, which underscores the connection between their respective formulations.
> > >
> > > **b. Advantages of Our Method Over Existing PLL Approaches**
> > >
> > > Existing PLL methods typically rely on disambiguation strategies that aim to refine the candidate label set by reducing it through mechanisms such as confidence scores or feature-based models. In contrast, our approach introduces a novel perspective through the lens of CLL. Specifically:
> > >
> > > **Positive Label Guessing (PLG)** component in our method effectively performs a thorough disambiguation by narrowing the candidate label set down to a single guessed true label.
> > >
> > > **Negative Label Enhancement (NLE)** mechanism, derived from the CLL framework, contributes to disambiguation by enhancing the representation of negative labels. This augmentation inherently reduces the ambiguity in the candidate label set, providing a complementary perspective to the traditional PLL disambiguation methods.
> > >
> > > Our method offers a fundamentally new perspective that previous PLL methods have not explored. This perspective is proved effective by our experimental results compared with SOTA PLL methods.
> > >
> > > **c. Reiteration of the Foundation and Contribution of Our Method**
> > >
> > > The method proposed in our paper is fundamentally built upon addressing the limitations of existing CLL methods instead of PLL methods. Our method introduces improvements that lead to state-of-the-art (SOTA) performance, as demonstrated through extensive experiments.
> > >
> > > We hope this response clarifies the contributions and novelty of our work, as well as advantages over existing PLL methods. Thank you again for your constructive comments.
> > >
> > > Best regards,
> > >
> > > Authors

---

> > > > ### Comment · Reviewer_zFpU · 2024-11-28
> > > >
> > > > Thanks for the response. I'd like to raise my score to 5. I still have the following concerns.
> > > > According to the responses, the advantages of the proposed PLG and NLG modules over current  PLL works remain unclear. On the one hand, the substantial innovation of the PLG module over most PLL disambiguation strategies remains unarticulated. On the other hand, the idea of the NLE module seems similar to that of the paper "N. Xu, et al. Variational Label Enhancement for Instance-Dependent Partial Label Learning. TPAMI".

---

> ### Author Response · Authors · 2024-11-28
>
> Thank you so much for raising your score. We deeply appreciate your constructive comments. Below, we address your remaining concerns in detail:
>
> **a. Key Differences Between PLNL and Common PLL Methods**
>
> We have conducted extensive literature review to distinguish our method from conventional PLL methods. Most existing PLL methods adopt a training-centric approach, where soft labels are employed to guide the **disambiguation process**. For example, PRODEN (ICML 2020), uses the predictive model's confidence scores to assign varying weights to candidate labels, guiding further training. In contrast, our method emphasizes the enhancement of the original data annotations, aiming to guide the model toward learning updated, hard-label information. This makes our approach inherently data-centric, rather than training-centric. By focusing on reinforcing the data itself, our method achieves **better interpretability** and more **robust learning outcomes**. For example, assume we have a complementary label instance whose label is [0, 0, 1, 0] (1 denote complementary label), after the PLG and NLE process, we changed the original data annotation into [0, 1, 1, 1]. This can be quickly interpreted as belonging to the first category. In contrast, the soft label information after disambiguation [0.36, 0.32, 0.07, 0.25] is very hard to explain. Although one may argue that hard-label information may bring more irrevocable errors for training, the error rate is well controlled with our double assurance in terms of method design and theoretical analysis.
>
> **b. Difference Between PLG and Disambiguation in Common PLL Methods**
>
> While both approaches leverage model outputs confidence, their applications differ significantly. As we mentioned in **a.**, common PLL methods employ soft label to guide the disambiguation process. In contrast, PLG adopts an instance-aware self-adaptive threshold, a two-view network and a well-designed set of criteria to directly identify the true label with high precision. PLG effectively classifies highly-confident samples as true-labeled samples, akin to labeled samples in semi-supervised learning. This, we have proved, eliminates ambiguity and provides a more reliable foundation for subsequent learning steps.
>
> **c. Comparison Between NLE and the Mentioned Paper**
>
> The referenced paper, "Variational Label Enhancement for Instance-Dependent Partial Label Learning" (TPAMI), utilizes a training-centric framework that models label distributions as probabilistic distributions updated iteratively. This process relies on variational Bayesian inference, incorporating sample features, label information, and neighborhood relationships. In contrast, our NLE module operates directly within the original label space. It builds upon the intuition that similar samples in the original label space share similar representations. Based on this insight, we introduce a complementary-label sharing mechanism that enriches the original label space with additional information. NLE supplements the dataset by adding complementary label information, offering a fundamentally different perspective.
>
> **d. Theoretical Contributions of PLNL**
>
> We emphasize that our theoretical contributions are novel and distinct within the PLL literature. We offer not only **practical advantages** (reflected in SOTA performance in almost all settings) but also the **interpretability and theoretical foundation** of CLL methodologies.
>
> We hope these clarifications address your concerns and further illustrate the distinctiveness and advantages of our proposed method. Thank you again for considering our response.

---

### Official Review · Reviewer_K5tE · 2024-10-31

**Soundness:** 3
**Presentation:** 3
**Contribution:** 2
**Rating:** 5
**Confidence:** 5

**Summary:**

This paper solved the complementary label learning problem by dividing the training instances into three categories: highly-confident, moderately-confident and under-confident. Then, it performed positive label guessing (PLG) and negative label enhancement (NLE) according to the categories of the instances. In addition, it unified PLG and NLE into a consistent framework. It also studied the bounds of the error rates of both PLG and NLE.

**Strengths:**

1. The study of generalization error bound is useful for algorithm evaluation.
2. The improvement in empirical study is significant.

**Weaknesses:**

1. There is no performance reported under biased complementary label learning settings.
2. There is no performance reported under manually annotated datasets, such as CLCIFAR10，CLCIFAR20. In addition, the results on these manually annotated data indicates the performance under instance-depedent CLL.
3. The semi-supervised learning methods used in this paper is not the SoTA ones.
4. There is no deviation reported in Table 3 and Table 4. It is suggested to add standard deviations or confidence intervals for the results, which would help readers better assess the statistical significance and reliability of the reported performance.
5. The study of the separation of moderately-confident and under-confident is not reported in the ablation. It will be helpful for evaluate the idea by comparing the performance when treating these two categories the same versus separately.

**Questions:**

1. Please consider the point 1 and 2 in the weaknesses.
2. How to evaluate the contribution of the separation of moderately-confident and under-confident instances?
3. Assumption 1 should be reconsidered and carefully checked. It assume that the variety of negative labels is large enough in the KNN instances. However, I think this assumption is too strict.
4. Is there any difference between the supervised and complementary learning losses for the high-confident instances? Maybe there are different from the loss gradient perspective.
5. The negative label set size in eq.(15) is fixed at $\tau_i$. However, when the top-$\tau_i$ and the complementary label is largely overlapped in the multiple complementary label setting, the benefit from the complementary label sharing mechanism is largely weakened.

---

> ### Author Response · Authors · 2024-11-22
> **Response to Reviewer K5tE PART 1**
>
> Thank you so much for your insightful comments!
>
> **W1: There is no performance reported under biased complementary label learning settings.**
>
> A: Thank you for pointing out this issue! We add the experimental results of three datasets (STL-10, CIFAR-10, CIFAR-100) under biased complementary label learning settings.
>
> To generate biased complementary labels, we first give the probability of each complementary label to be selected, following the same settings in [1], where two settings are given, including "without 0" and "with 0".
>
> In the “without0” setting, for each class $y$, we first randomly split $Y \textbackslash\{y\}$ to three subsets, each containing three elements. Then, for each complementary label in these three subsets, the probabilities are set to $\frac{0.6}{3}$ , $\frac{0.3}{3}$ , and $\frac{0.1}{3}$,  respectively.
>
> In the “with0” setting, for each class $y$, we first randomly selected three labels in $Y \textbackslash\{y\}$, and then randomly assign them with three probabilities whose summation is 1.
>
> **Without 0:**
>
> | Method  | STL-10     | CIFAR-10   | CIFAR-100 |
> | ------- | ---------- | ---------- | --------- |
> | UB-EXP  | 52.43%     | 22.54%     | **1.18%** |
> | UB-LOG  | 48.36%     | 22.71%     | 1.00%     |
> | SCL-EXP | 60.85%     | 23.53%     | 1.07%     |
> | SCL-LOG | 58.63%     | 22.96%     | 0.93%     |
> | POCR    | 74.78%     | 29.03%     | 1.00%     |
> | PLNL    | **75.38%** | **38.82%** | 1.00%     |
>
> **With 0:**
>
> | Method  | STL-10     | CIFAR-10   | CIFAR-100 |
> | ------- | ---------- | ---------- | --------- |
> | UB-EXP  | 60.71%     | 78.73%     | 1.00%     |
> | UB-LOG  | 44.42%     | 78.69%     | 1.00%     |
> | SCL-EXP | 61.24%     | 79.17%     | 1.00%     |
> | SCL-LOG | 63.36%     | 81.51%     | 1.00%     |
> | POCR    | 74.69%     | 94.71%     | 1.00%     |
> | PLNL    | **74.88%** | **95.15%** | 1.00%     |
>
> The results reveal two main points:
>
> 1. The advantage of PLNL over previous CLL methods still exists in biased settings, which demonstrates the robustness of PLNL. For example, under CIFAR-10 "without 0", PLNL outperforms previous SOTA by $9.79\%$.
> 2. All CLL methods cannot converge on biased CIFAR-100.
>
> **W2: There is no performance reported under manually annotated datasets, such as CLCIFAR10，CLCIFAR20. In addition, the results on these manually annotated data indicates the performance under instance-depedent CLL.**
>
> A: Thank you for pointing out this issue!
>
> We add the experimental results of CLCIFAR-10 and CLCIFAR-20 [2]. The results are as follows.
>
> | Method  | CLCIFAR-10 | CLCIFAR-20 |
> | ------- | ---------- | ---------- |
> | UB-EXP  | 41.95%     | 8.08%      |
> | UB-LOG  | 38.46%     | 6.00%      |
> | SCL-EXP | 40.93%     | 7.86%      |
> | SCL-LOG | 38.77%     | 7.45%      |
> | POCR    | 51.83%     | 8.82%      |
> | SELF-CL | 44.35%     | 8.64%      |
> | ComCo   | 46.42%     | 8.21%      |
> | PLNL    | **64.39%** | **9.88%**  |
>
> It can be seen from the above table that PLNL performs much better than previous CLL methods in real-world scenarios, with a $12.56\%$ improvement on CLCIFAR-10 and $1.06\%$ improvement on CLCIFAR-20 compared to previous SOTA (POCR).
>
> **W3: The semi-supervised learning methods used in this paper is not the SoTA ones.**
>
> A: Thank you for asking this question! We have reviewed a large number of papers in the field of semi-supervised learning and found that Freematch (2022) is still the best semi-supervised learning method in many visual datasets, including CIFAR100, etc.
>
> In order to ensure the diversity of the compared methods, we added the experimental results of the SemiReward [3] (ICLR 2024) on CIFAR10. We updated the curves pictures in **Appendix G.5.** for your reference. It can be seen that PLNL still outperforms SemiReward significantly in selection ratio and average size of NLS.
>
> **W4: There is no deviation reported in Table 3 and Table 4. It is suggested to add standard deviations or confidence intervals for the results, which would help readers better assess the statistical significance and reliability of the reported performance.**
>
> A: Thanks for pointing out this issue! We re-tested and added deviation terms in Table 3 and Table 4 of the paper. For example, on STL10 (SCLL), PLNL achieves $55.25\pm0.36\%$, PLNL v1 achieves $49.29 \pm 0.41\%$, PLNL v2 achieves $49.79\pm0.39\%$, PLNL  v3 achieves $53.25\pm0.35\%$, etc. The detailed results are added to **Appendix H** in the end of the paper.

---

> ### Author Response · Authors · 2024-11-22
> **Response to Reviewer K5tE PART 2**
>
> **W5: The study of the separation of moderately-confident and under-confident is not reported in the ablation. It will be helpful for evaluate the idea by comparing the performance when treating these two categories the same versus separately. How to evaluate the contribution of the separation of moderately-confident and under-confident instances?**
>
> A: Thank you for pointing out this issue!
>
> In our method, dividing the dataset into three parts is much natural than two parts. Since we train a two-view network for instance selection, it naturally divides the whole dataset into three parts based on the number of the output confidences of the two networks meeting the selection criteria:
>
> - Highly-confident: both of two confidences meet the criteria
>
> - Moderately-confident: one of the two confidences meets the criteria
>
> - Under-confident: both two confidences do not meet the criteria
>
> Since the model is under-learned for under-confident instances, the feature space of under-confident instances might be inaccurate, performing NLE based on this would lead to considerable errors. We show the NLE error rate of two strategies in the following table. **'PLNL w/ same strategy'** denotes treating moderately-confident and under-confident instances the same.
>
> | Method                | CIFAR-10 (SCLL)<br />NLE error rate $\epsilon_2$ | CIFAR-10 (MCLL)<br />NLE error rate $\epsilon_2$ | CIFAR-100 (MCLL)<br />NLE error rate $\epsilon_2$ |
> | --------------------- | ------------------------------------------------ | ------------------------------------------------ | ------------------------------------------------- |
> | PLNL w/ same strategy | 3.42$\pm$0.06%                                   | 2.11$\pm$0.12%                                   | 25.36$\pm$0.09%                                   |
> | PLNL                  | **2.21$\pm$0.09%**                               | **1.09$\pm$0.07%**                               | **19.16$\pm$0.04%**                               |
>
> And the final classification accuracy increases significantly under PLNL strategy.
>
> | Method                | CIFAR-10 (SCLL)     | CIFAR-10 (MCLL)     | CIFAR-100 (MCLL)    |
> | --------------------- | ------------------- | ------------------- | ------------------- |
> | PLNL w/ same strategy | 93.98$\pm$0.15%     | 95.89$\pm$0.24%     | 62.25$\pm$0.34%     |
> | PLNL                  | **94.78$\pm$0.12%** | **96.80$\pm$0.28%** | **64.33$\pm$0.43%** |
>
> It can be seen from the above tables that the separate treatment strategy used by PLNL outperforms the same treatment strategy. Same results have been updated to the paper.
>
> **Q1: Please consider the point 1 and 2 in the weaknesses.**
>
> A1: Please refer to W1 and W2.
>
> **Q2: How to evaluate the contribution of the separation of moderately-confident and under-confident instances?**
>
> A2: Please refer to W5.
>
> **Q3: Assumption 1 should be reconsidered and carefully checked. It assume that the variety of negative labels is large enough in the KNN instances. However, I think this assumption is too strict.**
>
> A: Thank you for your thoughtful comments! We will address this issue both qualitatively and quantitatively.
>
> **a. Qualitatively, we make the assumption for the convenience of theoretical analysis framework.**
>
> Recall that in Assumption 1, we assume that the positive label $y_i$ exists in its $k$-NN instances' complementary label set with probability no more than $\alpha_k$, any negative label $y_i'\neq y_i$ exist in its $k$-NN instances' complementary label set $\bar{Y}_{i}^{(j)}$ with probability no less than $\beta_k$.
>
> Here, we make such an assumption mainly for the convenience of whole theoretical analysis framework. Qualitatively speaking, we can forsee that $\beta_k$ will be larger in uniform label distribution while $\beta_k$ will be smaller in highly biased label distribution where the complementary label sharing mechanism might be weakened.
>
> However, no matter what the value of $\beta_k$ is, we can get Theorem 2 under such assumption, which strictly bounded the error rate of NLE. In the whole process, we did not restrict the value of $\beta_k$, so this assumption is not as strict as we might imagine.
>
> **b. Quantitatively, not only NLE error rate is very small across all cases, but PLNL still outperforms all previous CLL methods under biased CLL settings.**
>
> For example, for CIFAR-10 (SCLL), we showed that the the NLE precision reaches about 2.21%. Meanwhile, for CIFAR-10 ("without 0"), PLNL outperforms previous SOTA by $9.79\%$. This phenomenon again prove that our assumption is realistic and robust across all cases.

---

> ### Author Response · Authors · 2024-11-22
> **Response to Reviewer K5tE PART 3**
>
> **Q4: Is there any difference between the supervised and complementary learning losses for the high-confident instances? Maybe there are different from the loss gradient perspective.**
>
> A: Thank you for asking this question! The conclusion is that the supervised cross entropy loss and the complementary label learning loss is exactly the same for highly-confident instances. Please see the derivation below.
>
> First, please recall the CLL loss formulation in the paper as follows.
>
> $$
> \mathcal{L}_{CLL}(f(\boldsymbol{x}), \widehat{\bar{Y}}_i)
> $$
>
> $$
> =\sum\nolimits_{y\notin \widehat{\bar{Y}}_i}\frac{1}{K-\widehat{|\bar{Y}}_i|}\ell(f(\boldsymbol{x}),y)
> $$
>
> Obviously, this loss works by increasing the network's output on non-complementary labels and thus naturally suppress the model's output on complementary labels (negative labels).
>
> For highly-confident instance, we've pointed out that the complementary label set $\widehat{\bar{Y}}_i$ contains all the remaining labels except the guessed positive label $y_i$.
>
> $$
> \widehat{\bar{Y}}_i=\{c|c\in Y_i, c\neq \hat{y}_i\}
> $$
>
> Then we have $\widehat{|\bar{Y}}_i|=K-1$, so the CLL loss is reduced to the following version.
>
> $$
> \mathcal{L}_{CLL}(f(\boldsymbol{x}), \widehat{\bar{Y}}_i)
> $$
>
> $$
> =\sum\nolimits_{y\notin \widehat{\bar{Y}}_i}\ell(f(\boldsymbol{x}),y)
> $$
> $$
> =\ell(f(\boldsymbol{x}),\hat{y}_i)
> $$
>
> $$
> =\mathcal{L}_{cross.entropy}(f(\boldsymbol{x}_i),\hat{y}_i)
> $$
>
> The purpose of using CLL loss instead of directly computing CE loss for highly-confident instances is to remain the same loss form for both moderately-confident and under-confident instances for the convenience of theoretical analysis.
>
> **Q5: The negative label set size in eq.(15) is fixed at τi. However, when the top-τi and the complementary label is largely overlapped in the multiple complementary label setting, the benefit from the complementary label sharing mechanism is largely weakened.**
>
> A: Thank you for your insightful comments! Actually, we agree with you on this point: the benefit from the complementary label sharing mechanism might be weakened under biased CLL settings. However, we sincerely recommend that you reconsider the advantages of PLNL from a global perspective.
>
> **a. PLNL still outperforms all the previous CLL methods under biased CLL settings.**
>
> Please refer to results in W1 above. The results under biased CLL settings clearly demonstrate the superiority of PLNL.
>
> **b. PLNL show robustness compared with previous CLL methods, with not much degradation under biased CLL settings.**
>
> We show the performance degradation of different CLL methods under biased settings "without 0" compared to uniform settings. The results actually show that PLNL is robust than previous SOTA CLL method POCR.
>
> | Method  | CIFAR-10<br />Uniform | CIFAR-10<br />Biased ('without 0') | Difference |
> | ------- | --------------------- | ---------------------------------- | ---------- |
> | UB-EXP  | 62.90%                | 22.54%                             | 40.36%     |
> | UB-LOG  | 70.28%                | 22.71%                             | 47.57%     |
> | SCL-EXP | 72.35%                | 23.53%                             | 48.82%     |
> | SCL-LOG | 79.87%                | 22.96%                             | 56.91%     |
> | POCR    | 94.15%                | 29.03%                             | 65.12%     |
> | PLNL    | **94.78%**                | **38.82%**                             | 55.96%     |
>
> **c.  We sincerely recommend that you reconsider the advantages of PLNL from a global perspective.**
>
> In conclusion, although the concern might exist, but PLNL is robust in most cases. This is not only due to NLE's complementary label sharing mechanism, but also thanks to PLG's special design and the whole framework of PLNL.
>
> **Reference**
>
> [1] Yu X, Liu T, Gong M, et al. Learning with biased complementary labels[C]//Proceedings of the European conference on computer vision (ECCV). 2018: 68-83.
>
> [2] Wang H H, Ha M T, Ye N X, et al. CLImage: Human-Annotated Datasets for Complementary-Label Learning[J]. 2024.
>
> [3] Li S, Jin W, Wang Z, et al. Semireward: A general reward model for semi-supervised learning[J]. arXiv preprint arXiv:2310.03013, 2023.

---

> > ### Comment · Reviewer_K5tE · 2024-11-24
> >
> > Thanks for your rebuttals. Some details of my concerns have been partly addressed. However, for the overall evaluation and the concern on the novelty, I need to further discuss with the SPC and other PCs.

---

> ### Author Response · Authors · 2024-11-25
>
> Thank you for your thoughful feedback. We would like to reclaim the key contributions of our work:
>
> - **Theoretical Analysis:** The introduction of theoretical generalization error bounds and error rate bounds to enhance the soundness and solidness of our method. It has never been revealed in the field of CLL before.
> - **Novel Methodology:** The novel approach of decomposing the complementary label learning (CLL) problem into positive label guessing (PLG) and negative label enhancement (NLE). We make the first attempt to construct a reasonable and comprehensive pseudo-labeling framework with theoretical foundations.
> - **SOTA Empirical Results:** PLNL reaches SOTA in CLL and its robustness has been proved in biased CLL, real-world scenarios, etc.
>
> Hopefully, this reiteration could serve as a probable solution to your concerns about the novelty of our work.

---

### Official Review · Reviewer_Gske · 2024-11-03

**Soundness:** 3
**Presentation:** 3
**Contribution:** 2
**Rating:** 5
**Confidence:** 3

**Summary:**

The paper introduces PLNL, a new method for complementary label learning that leverages positive label guessing and negative label enhancement based on a confidence-based instances selection module. Through a unified framework and theoretical analysis, it demonstrates the effectiveness of PLNL, achieving state-of-the-art results in CLL.

**Strengths:**

1. This paper is written clear and easy to understand.

2. This paper proposes a new method PLNL for CLL by PLG and NLE.

3. Extensive experiments demonstrate the effectiveness of PLNL over the SOTA CLL methods.

**Weaknesses:**

The contribution of this paper is incremental and using a commonly used confidence based instance selection strategy. Moreover,  only one SOTA method (published in 2023 or 2024) is used and it is too weak to validate the effectiveness of the proposed method.

**Questions:**

1. In the ablation study, it appears that the weak-strong data augmentation strategy is more effective than the proposed confidence-based instance selection strategy.

2. The settings outlined in Table 1 are lacking experimental results for the CIFAR100 dataset.

3.  The number of SOTA methods employed in the experiments is insufficient: only one SOTA method (published in 2023 or 2024) is used, which is inadequate and too weak to validate the effectiveness of the proposed method.

---

> ### Author Response · Authors · 2024-11-22
> **Response to Reviewer Gske PART 1**
>
> Thank you so much for your insightful comments!
>
> **W1: The contribution of this paper is incremental and using a commonly used confidence based instance selection strategy.**
>
> A: Thanks for raising concerns about this issue! We will reclaim the novelty of PLNL in four parts.
>
> **a. PLNL is well-motivated.**
>
> Previous studies on CLL can be roughly divided into two categories: methods that attempt to construct
> an unbiased risk estimator (URE-based) (Ishida et al., 2017; 2019; Feng et al., 2020) and methods
> based on feature learning (FL-based) (Chou et al., 2020; Wang et al., 2021; Liu et al., 2022; Jiang
> et al., 2024). These methods either focus on the design of robust loss functions or the exploration of representation learning, while neglecting the power of **output space information**, namely **confidence information**.
>
> Previous studies in semi-supervied learning (SSL) have already paved the way for confidence-based sample selection methods. However, previous SSL methods lack **the utilization of untrustworthy instances**. They either discard this part or simply employ techniques such as consistency regularization.
>
> We make the first attempt to **leverage the output space information (PLG)** to facilitate CLL, combining with **well-designed utilization of untrustworthy instances (NLE)**, to make up the whole framework of PLNL. Last but not least, we unify the pseudo-labeling methods in a negative label recovery framework and provide **theoretical analysis of PLNL**.
>
> **b. PLG is more than a simple confidence-based instance selection strategy.**
>
> 1. Instance-aware self-adaptive threshold. A commonly used single global threshold does not consider the fitting difficulties of different instances, i.e., hard instances and easy instances. Therefore, we design an instance-aware self-adaptive threshold
>    for each instance and update it iteratively.
>
> 2. Well-designed selection criteria. The three selection criteria gurantee that the confidence of highly-confident instances stable, high and steer clear of complementary labels.
>
> 3. Two-view networks. Inspired by ensemble learning, we train a two-view network for instance selection. It naturally divides the whole dataset into three parts based on whether the output confidences of the two meet the designed criteria:
>
>    - Highly-confident: both confidences meet the criteria
>
>    - Moderately-confident: one of the confidences meets the criteria
>
>    - Under-confident: both confidences do not meet the criteria
>
> **c. NLE deserves more attention in its feature-based complementary label sharing mechanism.**
>
> On the other hand, NLE is a feature-based label enhancement strategy. We calculate the $k$-NN instances at the feature level and enhance the negative labels based on the label information of the its $k$-NN instances. This, to the best of our knowledge, is the first attempt to leverage feature-level information in CLL literature.
>
> **d. Theoretical Analysis strengthen the soundness and solidness of PLNL**
>
> Last but not least, we argue that our theoretical contribution to CLL cannot be ignored. We unify PLG and NLE into the framework of negative label recovery and theoretically prove that the generalization error of PLNL is bounded when error rates of PLG and NLE are both close to zero. Furthermore, under Assumption 1 and Assumption 2, we also prove that the PLG and NLE error rates are both upper bounded by a small number close to zero. This double bounded the generalization of PLNL.
>
> **We sincerely recommend that you reconsider the advantages of PLNL from a global perspective.**
>
> **W2: Moreover, only one SOTA method (published in 2023 or 2024) is used and it is too weak to validate the effectiveness of the proposed method.**
>
> A: Thanks for pointing out this issue! We supplyment two SOTA partial label learning (PLL) methods: DPLL (ICML 2023) [1] and PiCO+(TPAMI 2024) [2] for comparison.
>
> | Method | CIFAR-10<br />SCLL  | CIFAR-10<br />MCLL  | CIFAR-100<br />SCLL | CIFAR-100<br />MCLL |
> | ------ | ------------------- | ------------------- | ------------------- | ------------------- |
> | DPLL   | 93.98$\pm$0.09%     | 95.49$\pm$0.11%     | 1.00%               | 61.47$\pm$0.21%     |
> | PiCO+  | 94.43%$\pm$0.07%    | 95.35$\pm$0.08%     | 1.00%               | 61.25$\pm$0.32%     |
> | PLNL   | **94.78$\pm$0.12%** | **96.80$\pm$0.28%** | 1.00%               | **64.33$\pm$0.43%** |
>
> It can be seen from the results that PLNL outperforms PLL SOTAs in all test cases.

---

> ### Author Response · Authors · 2024-11-22
> **Response to Reviewer Gske PART 2**
>
> **Q1: In the ablation study, it appears that the weak-strong data augmentation strategy is more effective than the proposed confidence-based instance selection strategy.**
>
> A: Thank you for your detailed observation. The confidence-based instance selection strategy is in fact more effective than the weak-strong data augmentation strategy. We will clear up some misunderstandings about the contents of the table in ablation study section.
>
> Table 3 reports the selection ratio of PLG $\eta$ and PLG precision $1-\epsilon_1$ with or without weak-strong data augmentation strategy. Table 4 (line 2 and line 3) reports the classification accuracy with or without weak-strong data augmentation strategy.
>
> Table 3 does not report the classification accuracy, please refer to Table 4 for further information.
>
> It can be seen from the result that the proposed confidence-based instance selection strategy (PLG) (line4 of Table 4) brings about more performance gain than data augmentation (line 3 of Table 4), $2.77\%$ compared to $1.03\%$ on CIFAR-10 (SCLL), etc.
>
> **Q2: The settings outlined in Table 1 are lacking experimental results for the CIFAR100 dataset.**
>
> A: Thank you for raising concerns about this issue! On CIFAR-100 in single complementary label setting, none of the CLL methods currently, including PLNL, can achieve significantly more than $1\%$ classification accuracy. Since the results are not discriminative, we only show CIFAR100 with multiple CLs.This scenorio is much challenging and need more future work.
>
> We've already tested the performance of PLNL on STL-10, SVHN, FMNIST, CIFAR10, CIFAR-100. Here, to respond to your concerns, we add the results on Tiny-ImageNet, CLCIFAR10, CLCIFAR20 for further validation.
>
> For complex benchmark dataset, the appendix already covered Tiny-ImageNet with multiple complementary labels. The dataset has 200 categories and at least 224 $\times$ 224  image size. The results are as follows.
>
> | Method  | Tiny-ImageNet |
> | ------- | ------------- |
> | UB-EXP  | 3.89%         |
> | UB-LOG  | 7.17%         |
> | SCL-EXP | 3.36%         |
> | SCL-LOG | 8.96%         |
> | POCR    | 4.29%         |
> | SELF-CL | 7.87%         |
> | ComCo   | 8.52%         |
> | PLNL    | **11.87%**    |
>
> It can be seen from the above that although most of the methods perform poorly, PLNL still outperforms traditional CLL methods significantly, with $2.91\%$ improvement compared to previous SOTA (SCL-LOG).
>
> For real-world scenarios, we added two datasets, including CLCIFAR10 and CLCIFAR20 [3], which are two manually-annotated datasets specially designed for CLL.  The results are as follows.
>
> | Method  | CLCIFAR-10 | CLCIFAR-20 |
> | ------- | ---------- | ---------- |
> | UB-EXP  | 41.95%     | 8.08%      |
> | UB-LOG  | 38.46%     | 6.00%      |
> | SCL-EXP | 40.93%     | 7.86%      |
> | SCL-LOG | 38.77%     | 7.45%      |
> | POCR    | 51.83%     | 8.82%      |
> | SELF-CL | 44.35%     | 8.64%      |
> | ComCo   | 46.42%     | 8.21%      |
> | PLNL    | **64.39%** | **9.88%**  |
>
> It can be seen from the above that PLNL perform much better than previous CLL methods in real-world scenarios, with a $12.56\%$ improvement on CLCIFAR-10 and $1.06\%$ improvement on CLCIFAR-20 compared to previous SOTA (POCR).
>
> The additional three datasets can be considered much harder scenorios than listed benchmark dataset in the main paper. PLNL is still SOTA on all of them.
>
> **Q3: The number of SOTA methods employed in the experiments is insufficient: only one SOTA method (published in 2023 or 2024) is used, which is inadequate and too weak to validate the effectiveness of the proposed method.**
>
> A: Please refer to W2.
>
> **Reference**
>
> [1] Wu D D, Wang D B, Zhang M L. Revisiting consistency regularization for deep partial label learning[C]//International conference on machine learning. PMLR, 2022: 24212-24225.
>
> [2] Wang H, Xiao R, Li Y, et al. Pico+: Contrastive label disambiguation for robust partial label learning[J]. IEEE Transactions on Pattern Analysis and Machine Intelligence, 2023.
>
> [3] Wang H H, Ha M T, Ye N X, et al. CLImage: Human-Annotated Datasets for Complementary-Label Learning[J]. 2024.

---

> > ### Comment · Reviewer_Gske · 2024-11-26
> > **Reply to the authors feedback**
> >
> > Thanks for the authors' response.
> >
> > It seems that the proposed method performs well on datasets with a small label space. However, I suspect that the SOTA method might outperform the proposed method on datasets with larger label sapce, such as CLCIFAR-30 or CLCIFAR-40.

---

> ### Author Response · Authors · 2024-11-25
>
> Dear reviewer Gske,
>
> Thanks again for your time and efforts in reviewing this paper and the valuable comments on improving its quality. As the reviewer-author discussion deadline approaches, please take a few minutes to read the rebuttal. If you have further concerns, we are happy to provide more explanations. Thanks.
>
> Regards from the authors.

---

> ### Author Response · Authors · 2024-11-26
>
> Thank you for your thoughtful response. We would like to address your concerns as follows:
>
> **a. Performance of PLNL on Larger Label Spaces**
>
> Our proposed method, PLNL, has demonstrated state-of-the-art performance on both the CIFAR-100 and Tiny-ImageNet datasets. The results are shown in **Table 2** and **Table 9** of the paper respectively. These datasets possess sufficiently large label spaces (200 and 100 classes, respectively), making them representative benchmarks for evaluating the effectiveness of methods under scenarios with extensive label spaces. For example, on CIFAR-100, PLNL outperforms previous SOTA by **6.45%** which is a remarkable performance gain.
>
> **b. Clarification Regarding CLCIFAR-30 and CLCIFAR-40**
>
> We would like to clarify that the datasets CLCIFAR-30 and CLCIFAR-40 do not exist. In the referenced paper [1], only CLCIFAR10, CLCIFAR20, CLMicroImageNet10 and CLMicroImageNet20 are collected as real-world datasets for complementary-label learning research. There is no mention of CLCIFAR-30 or CLCIFAR-40 in this work or elsewhere.
>
> We hope this clarification addresses your concerns and provides sufficient evidence of the robustness of our method in scenarios involving larger label spaces. Thank you again for your insightful comments.
>
> Best regards,
>
> Authors
>
> **Reference**
>
> [1] Wang H H, Ha M T, Ye N X, et al. CLImage: Human-Annotated Datasets for Complementary-Label Learning[J]. 2024.

---

> > ### Comment · Reviewer_Gske · 2024-12-02
> > **Reply to the authors feedback**
> >
> > Thanks for the authors' response.

---

### Official Review · Reviewer_K6ZB · 2024-11-03

**Soundness:** 3
**Presentation:** 2
**Contribution:** 2
**Rating:** 5
**Confidence:** 4

**Summary:**

This paper proposes a new Complementary Label Learning (CLL) method called PLNL (Positive Label Guessing and Negative Label Enhancement), which solves the CLL problem by decomposing the inverse problem into two subtasks, positive label guessing (PLG) and negative label enhancement (NLE). Specifically, PLNL classifies training instances into three categories based on the confidence evaluation of the model output: high confidence, medium confidence, and low confidence, and performs PLG and NNE on them respectively. This paper also proposes a unified framework that considers PLG and NLE as the process of negative label recovery, and theoretically proves that the error rates of PLG and NLE have upper bounds, thus enabling the construction of a model consistent with classifiers learned using clean and complete labels.

**Strengths:**

1.This paper provides a novel solution for the CLL field by transforming the CLL problem into an inverse problem that outputs spatial information and decomposing it into two subtasks, PLG and NLE.
2.This paper also provides a theoretical upper bound proof of error rate, enhancing the credibility and effectiveness of the method.

**Weaknesses:**

1. The performance of PLNL may be sensitive to parameter selection, such as k-NN parameter k and confidence threshold λ, which requires users to make detailed parameter adjustments when using it.
2.Although PLNL has performed well in experiments, its generalization ability on different datasets and tasks of different complexities still needs further validation.
3.The k-NN computation involved in PLNL may have high computational costs on large-scale datasets, which may limit its application in resource-constrained environments.

**Questions:**

Please see the weaknesses above.

---

> ### Author Response · Authors · 2024-11-22
> **Response to Reviewer K6ZB PART 1**
>
> Thank you so much for your insightful comments!
>
> **W1: The performance of PLNL may be sensitive to parameter selection, such as k-NN parameter k and confidence threshold λ, which requires users to make detailed parameter adjustments when using it.**
>
> A: Thank you for raising concerns about this issue! We have already run the hyperparameter sensitivity analysis, including experiments with various **memory bank sizes $t$, confidence threshold $\lambda$, as well as $k$-NN parameter $k$.** The results were provided in the **Appendix G.2**. We copy the results of $k$ hereby.
>
> #### $k$:
>
> | $k$  | STL-10<br />SCLL | CIFAR-10<br />SCLL | CIFAR-100<br />MCLL |
> | ---- | ---------------- | ------------------ | ------------------- |
> | 100  | **54.45%**       | 94.25%             | 63.27%              |
> | 250  | 55.25%           | 94.73%             | 64.33%              |
> | 500  | 47.23%           | **94.78%**         | **65.40%**          |
> | 1000 | 46.40%           | 94.65%             | 65.23%              |
>
> As shown in the table, the influence of $k$ value on the performance of PLNL is little on CIFAR-10 and CIFAR-100. When we choose k from [100, 250, 500, 1000], the performance fluctuation does not exceed $0.7\%$ on CIFAR-10 and $1.94\%$ on CIFAR-100.
>
> However, it is worth mentioning that a too large $k$ value (e.g. 1000) will indeed cause performance degradation on STL-10.
>
> This, in our opinion, results from the scarcity of information in dataset STL-10. STL-10 contains only 500 images per category. A too large $k$ value (e.g. 1000) which is larger than 500 will cause that a certain number of $k$-NN instances do not belong to the same class, hence resulting in much NLE errors and influencing the performance of PLNL.
>
> #### $\lambda$:
>
> In our work, PLNL employs a self-adaptive confidence threshold without manual tuning and achieves great performance.
> $$
> \lambda(t)=\alpha\lambda(t-1) + (1-\alpha)f(t),\lambda(0)=\frac{1}{K},
> $$
> In conclusion, in most cases, hyper-parameters have little to do with the classification performance of PLNL and hence no detailed manual tuning of hyper-parameters is required.
>
> **W2: Although PLNL has performed well in experiments, its generalization ability on different datasets and tasks of different complexities still needs further validation.**
>
> A: Thank you for your valuable recommendations! We've already tested the performance of PLNL on STL-10, SVHN, FMNIST, CIFAR10, CIFAR-100. Here, to respond to your concerns, we add the results on Tiny-ImageNet, CLCIFAR10, CLCIFAR20 for further validation.
>
> For complex benchmark dataset, the appendix already covered Tiny-ImageNet with multiple complementary labels. The dataset has 200 categories and at least 224 $\times$ 224  image size. The results are as follows.
>
> | Method  | Tiny-ImageNet |
> | ------- | ------------- |
> | UB-EXP  | 3.89%         |
> | UB-LOG  | 7.17%         |
> | SCL-EXP | 3.36%         |
> | SCL-LOG | 8.96%         |
> | POCR    | 4.29%         |
> | SELF-CL | 7.87%         |
> | ComCo   | 8.52%         |
> | PLNL    | **11.87%**    |
>
> It can be seen from the above table that although most of the methods performs poorly, PLNL still outperforms traditional CLL methods significantly, with $2.91\%$ improvement compared to previous SOTA (SCL-LOG).
>
> For real-world scenarios, we added two datasets, including CLCIFAR10 and CLCIFAR20 [1], which are two manually-annotated datasets specially designed for CLL.  The results are as follows.
>
> | Method  | CLCIFAR-10 | CLCIFAR-20 |
> | ------- | ---------- | ---------- |
> | UB-EXP  | 41.95%     | 8.08%      |
> | UB-LOG  | 38.46%     | 6.00%      |
> | SCL-EXP | 40.93%     | 7.86%      |
> | SCL-LOG | 38.77%     | 7.45%      |
> | POCR    | 51.83%     | 8.82%      |
> | SELF-CL | 44.35%     | 8.64%      |
> | ComCo   | 46.42%     | 8.21%      |
> | PLNL    | **64.39%** | **9.88%**  |
>
> It can be seen from the above table that PLNL performs much better than previous CLL methods in real-world scenarios, with a $12.56\%$ improvement on CLCIFAR-10 and $1.06\%$ improvement on CLCIFAR-20 compared to previous SOTA (POCR).
>
> The additional three datasets can be considered much harder scenorios than listed benchmark dataset in the main paper. PLNL is still SOTA on all of them.

---

> > ### Author Response · Authors · 2024-11-22
> > **Response to Reviewer K6ZB PART 2**
> >
> > **W3: The k-NN computation involved in PLNL may have high computational costs on large-scale datasets, which may limit its application in resource-constrained environments.**
> >
> > A: Thank you for raising concerns about this issue!  We count the running time of PLNL and baseline methods. We tested the running time on CIFAR-100 (MCLL), each method runs on an empty NVIDIA RTX 3090 without any disturbance.
> >
> > | Method  | Running Time | Accuracy   |
> > | ------- | ------------ | ---------- |
> > | POCR    | **3h15m**    | 53.16%     |
> > | SELF-CL | 8h24m        | 57.65%     |
> > | ComCo   | 15h34m       | 57.88%     |
> > | PLNL    | 12h42m       | **65.40%** |
> >
> > From the results above, we can find that although PLNL takes more running time compared with some baseline methods, it is faster than ComCo. However, the performance gain is much, for example a $12.24\%$ performance gain compared to POCR.
> >
> > **Reference**
> >
> > [1] Wang H H, Ha M T, Ye N X, et al. CLImage: Human-Annotated Datasets for Complementary-Label Learning[J]. 2024.

---

> ### Author Response · Authors · 2024-11-25
>
> Dear reviewer K6ZB,
>
> Thanks again for your time and efforts in reviewing this paper and the valuable comments on improving its quality. As the reviewer-author discussion deadline approaches, please take a few minutes to read the rebuttal. If you have further concerns, we are happy to provide more explanations. Thanks.
>
> Regards from the authors.

---

> > ### Author Response · Authors · 2024-11-27
> >
> > Dear reviewer K6ZB,
> >
> > Thank you once again for your time and valuable feedback on our paper. As the deadline for submitting the final PDF is now less than 24 hours away, we kindly remind you to review the rebuttal and let us know if you have any further questions or concerns. We would be happy to provide additional clarifications if needed.
> >
> > We greatly appreciate your support and timely response.
> >
> > Best regards,
> >
> > The authors

---

### Official Review · Reviewer_ocgG · 2024-11-04

**Soundness:** 3
**Presentation:** 3
**Contribution:** 2
**Rating:** 6
**Confidence:** 3

**Summary:**

In this paper, the authors address the complementary label learning problem by performing positive label guessing (PLG) and negative label enhancement (NLE). The authors prove the generalization error bound and the error rates bounds. Extensive experiments demonstrate the superiority of the proposed method.

**Strengths:**

1.	The authors provide theoretical generalization error bound for their proposed method.
2.	The results of the experiments verify the effectiveness of the method.
3.	This paper is well-organized and easy to understand.

**Weaknesses:**

1.	According to Eq.(21), the upper bound of the PLG error rate ranges from 0 to 1, which is too broad and meaningless. And it is not easy to estimate the interval of the error bound in Eq.(22). It would be more meaningful if the authors could show that the upper bounds for these error rates are small numbers close to zero.
2.	Most datasets used in Tables 1 and 2 are relatively easy, it would be better if data set such as 20 Newsgroups is included for comparison.
3.	Some of the experimental results need further explanation, please refer to the questions below.

**Questions:**

1.	In Figure 2(b), the PLG precision decreases as the training progresses, while the NLE precision remains relatively stable, please explain this.
2.	Why do you report the method POCR in Table 4?
3.	Please give the detailed derivation of Eq.(30), especially the first inequality.
4.	There are some typos, e.g., in Eq.(32), “B” is missing, in line 134, “show” should be “shown”, in line 169, ”p” should be “f”, in line 216, “are” should be “is”, and in Figures 2(b) and 2(c), “Precison” should be “Precision”.

---

> ### Author Response · Authors · 2024-11-22
> **Response to Reviewer ocgG PART 1**
>
> Thank you so much for your insightful comments!
>
> **W1: According to Eq.(21), the upper bound of the PLG error rate ranges from 0 to 1, which is too broad and meaningless. And it is not easy to estimate the interval of the error bound in Eq.(22). It would be more meaningful if the authors could show that the upper bounds for these error rates are small numbers close to zero.**
>
> A: Thanks for pointing out this issue! We'll elucidate both the error rates of PLG and NLE empirically and theoretically as follows.
>
> **a. Empirical results show that PLG/NLE error rate are low**
>
> Empirically, in fact, we have already showed that both the error rates of PLG and NLE are close to zero. Please refer to Table 5 in Appendix for further information. For example, for CIFAR-10 (SCLL), we showed that the the precision of PLG and NLE reach 96.79$\pm$0.08% and 97.79$\pm$0.09% respectively, which in turn gives very low error rates of approximately 3.21% and 2.21%.
>
> **b. Theoretical results show that the upper bounds for these error rates are small numbers much close to zero.**
>
> **Theorem3:** Let's first take a look at the right side of Eq.(22) of the paper.
> $$
> \sum_{j=1}^k \binom{|Y_i| - 1}{|Y_i| - \tau_i} F_{\beta_k}(k - j + 1, j)^{(|Y_i| - \tau_i)} b_{\alpha_k}(k, j)
> $$
>
> where:
>
> - $\binom{|Y_i| - 1}{|Y_i| - \tau_i}$ is a bin binomial coefficient, which is a non-negative integer, and in this context, it scales the term.
>
> - $F_{\beta_k}(k(k - j + 1, j)$ $)$ is the regularized incomplete beta function, which lies in the range $[0, 1]$.
>
> - $b_{\alpha_k}(k, j)$ represents the probability mass function of a binomial distribution $B(k, \alpha_k)$, also within $[0, 1]$.
>
>
> Since $F_{\beta_k}(k - j + 1, j)$ is within $[0, 1]$, raising it to a high power, specifically $ (|Y_i| - \tau_i) $, which is assumed to be large in practical scenarios, causes this term to approach zero, making the contribution of each term in the sum negligible.
>
> Thus, the right-hand side of the inequality is approximately:
> $$
> \sum_{j=1}^k \binom{|Y_i| - 1}{|Y_i| - \tau_i} F_{\beta_k}(k - j + 1, j)^{(|Y_i| - \tau_i)} b_{\alpha_k}(k, j) \approx 0
> $$
>
> **Theorem2:** Though the theoretical approximation of $\epsilon_1$ is hard, let's look at Eq.(21) and its conditions.
> $$
> \epsilon_1=\mathbb{P}(y_i \in \widehat{\bar{Y}}_i)\le (K-1-s_i)\psi,
> $$
>
> where $K$ is class number, $s_i$ is the size of $\bar{Y}_i$ and $\psi\in (0,\frac{1}{K-1-s_i})$.
>
> Note that $(0,\frac{1}{K-1-s_i})$ is an open interval so that the worst case, where $\epsilon_1=1$ will always be avoided. From this point, Theorem 3 indicates that performing PLE is not a bad choice.
>
> **W2: Most datasets used in Tables 1 and 2 are relatively easy, it would be better if data set such as 20 Newsgroups is included for comparison.**
>
> A: Thank you for your valuable recommendations! We are sorry to say that newest CLL methods usually rely on weak and strong augmentations on images, which cannot be used identically on text datasets such as 20Newsgroups.
>
> However, to respond to your concerns, we add more experimental results on TinyImageNet, CLCIFAR-10, CLCIFAR-20.
>
> For complex benchmark dataset, the appendix already covered TinyImageNet-200 with multiple complementary labels. The dataset has 200 categories and at least 224 $\times$ 224  image size. The results are as follows.
>
> | Method  | Tiny-ImageNet |
> | ------- | ------------- |
> | UB-EXP  | 3.89%         |
> | UB-LOG  | 7.17%         |
> | SCL-EXP | 3.36%         |
> | SCL-LOG | 8.96%         |
> | POCR    | 4.29%         |
> | SELF-CL | 7.87%         |
> | ComCo   | 8.52%         |
> | PLNL    | **11.87%**    |
>
> It can be seen from the above table that although most of the methods perform poorly, PLNL still outperforms traditional CLL methods significantly, with $2.91\%$ improvement compared to previous SOTA (SCL-LOG).
>
> For real-world scenarios, we added two datasets, including CLCIFAR10 and CLCIFAR20 [1], which are two manually-annotated datasets specially designed for CLL.  The results are as follows.
>
> | Method  | CLCIFAR-10 | CLCIFAR-20 |
> | ------- | ---------- | ---------- |
> | UB-EXP  | 41.95%     | 8.08%      |
> | UB-LOG  | 38.46%     | 6.00%      |
> | SCL-EXP | 40.93%     | 7.86%      |
> | SCL-LOG | 38.77%     | 7.45%      |
> | POCR    | 51.83%     | 8.82%      |
> | SELF-CL | 44.35%     | 8.64%      |
> | ComCo   | 46.42%     | 8.21%      |
> | PLNL    | **64.39%** | **9.88%**  |
>
> It can be seen from the above table that PLNL performs much better than previous CLL methods in real-world scenarios, with a $12.56\%$ improvement on CLCIFAR-10 and $1.06\%$ improvement on CLCIFAR-20 compared to previous SOTA (POCR).
>
> The additional three datasets can be considered as much harder scenorios than listed benchmark dataset in the main paper. PLNL is still SOTA on all of them.

---

> ### Author Response · Authors · 2024-11-22
> **Response to Reviewer ocgG PART 2**
>
> **Q1: In Figure 2(b), the PLG precision decreases as the training progresses, while the NLE precision remains relatively stable, please explain this.**
>
> A: We will address this issue in two parts.
>
> **a. The reason why PLG precision decreases.**
>
> Confidence-based methods generally show a slight decrease in precision as the number of training epochs increases due to overfitting. (It's hard to have both high selection ratio $\eta$ and high precision $1-\epsilon_1$.) However, the selection ratio increases gradually and reaches a high value at last in Fig. 2(b). Since the performance is continuously on the increase, it is worthy of sacrificing little precision for much higher selection ratio. In other cases, the selection ratio is even higher with even less loss of precision. In our experiment, we achieved $93.40\%$ selection ratio with $97.79\%$ precision on CIFAR-10 (SCLL) and $97.49\%$ selection ratio with $98.91\%$ precision on CIFAR-10 (MCLL). In Fig. 2(b), we showed a $76.08\%$ selection ratio with $80.84\%$ precision on CIFAR-100 (MCLL), which significantly surpasses Fixmatch and Freematch. (Detailed reports in Table 5 in appendix)
>
> **b. The reason why NLE precision remains stable and high.**
>
> NLE precision remains stable and high mainly because of the divide-and-conquer strategy of moderately-confident and under-confident instances. Instead of treating all instances with low confidence equally, we utilize a strategy where the higher the confidence, the more enhanced negative labels will be generated.
>
> In our experiment, we achieved $93.40\%$ selection ratio with $97.79\%$ precision on CIFAR-10 (SCLL) and $97.49\%$ selection ratio with $98.91\%$ precision on CIFAR-10 (MCLL). In Fig. 2(b), we showed a $76.08\%$ selection ratio with $80.84\%$ precision on CIFAR-100 (MCLL), which significantly surpasses Fixmatch and Freematch.
>
> **Q2: Why do you report the method POCR in Table 4?**
>
> A:  POCR is the previous SOTA before PLNL, so we show results of POCR as a reference.
>
> **Q3: Please give the detailed derivation of Eq.(30), especially the first inequality.**
>
> A: Thank you for interest in theoretical analysis of PLNL! We add detailed information for each step of the derivation.
> Because that the edit box does not support the compilation of long formulas, the detailed derivation is added to **Appendix I** in the end of the paper. We tried our best to analyse each pace of the derivation and make sure it is clear and easy to understand.
>
> **Q4: There are some typos., e.g., in Eq.(32), “B” is missing, in line 134, “show” should be “shown”, in line 169, ”p” should be “f”, in line 216, “are” should be “is”, and in Figures 2(b) and 2(c), “Precison” should be “Precision”.**
>
> A: Thanks for your correction! We will try our best to check to ensure that similar typos will not occur.
>
> **Reference**
>
> [1] Wang H H, Ha M T, Ye N X, et al. CLImage: Human-Annotated Datasets for Complementary-Label Learning[J]. 2024.

---

> ### Author Response · Authors · 2024-11-25
>
> Dear reviewer ocgG,
>
> Thanks again for your time and efforts in reviewing this paper and the valuable comments on improving its quality. As the reviewer-author discussion deadline approaches, please take a few minutes to read the rebuttal. If you have further concerns, we are happy to provide more explanations. Thanks.
>
> Regards from the authors.

---

> > ### Comment · Reviewer_ocgG · 2024-11-25
> >
> > Thanks for your time and efforts in addressing my concerns, but I do not figure out how the $K-\bar{s} -1$ (in the numerator) of Eq.(51) comes from. Please explain it.

---

> > > ### Author Response · Authors · 2024-11-25
> > >
> > > Thanks for your reply. Since the edit box does not support the compilation of long formulas, in order to address your concerns, we have updated the **Appendix I** with extra details, especially Eq.(50) and Eq.(51) and their descriptions. The modified contents are marked in $\textcolor{red}{red}$ font.
> > >
> > > Hopefully this will address your concerns. If you have any further concerns, we are looking forward to more constructive discussions with you.

---

> > > > ### Comment · Reviewer_ocgG · 2024-11-26
> > > >
> > > > Thanks for your response. I currently have no further questions and will retain my positive score.

---

### Author Response · Authors · 2024-11-23

Dear Reviewers,

Greetings!

We sincerely thank all reviewers for their efforts and valuable feedback, which have been instrumental in improving the quality of our manuscript. We are delighted that the reviewers have recognized the key contributions of our work, including:

- **Theoretical Contributions:** The introduction of theoretical generalization error bounds and error rate bounds (noted by **all reviewers**) to enhance the soundness and solidness of our method.
- **Methodology:** The novel approach of decomposing the complementary label learning (CLL) problem into positive label guessing (PLG) and negative label enhancement (NLE), effectively addressing the challenges in CLL (highlighted by Reviewers **K6ZB, Gske, and zFpU**).
- **Empirical Results:** The robustness and effectiveness of our method, supported by extensive experiments that demonstrate superior performance compared to state-of-the-art methods (appreciated by **all reviewers**).
- **Presentation and Clarity:** The paper’s clear organization and accessibility, making the methodology and results easy to follow (acknowledged by Reviewers **ocgG, Gske, and zFpU**).

Our manuscript has been updated, and we will tackle the concerns of each reviewer point-by-point.

Thank you once again for your thoughtful and valuable feedback. We are looking forward to further constructive discussions with you!

Best regards,

Authors

---

### Meta-Review · Area_Chair_j63N · 2024-12-10

**Metareview:**

This paper proposes a new complementary label learning method called PLNL, which use positive label guessing and negative label enhancement. It is based on the confidence of the model output, and divide the training into 3 groups and design a way to use each group effectively with techniques such as pseudo labels. The paper also shows theoretical analysis by showing the error rates of both PLG and NLE are upper bounded. Experimental results show the superiority of the proposed method over previous complementary label learning methods.

Reviewers noted that the method is meaningful, and the theoretical contributions provide valuable insights. Reviewers acknowledged that experimental results demonstrate the performance gains of the proposed method, and felt the clarity and presentation of the paper is good. On the other hand, reviewers expressed concerns about bound tightness or assumptions/conditions used, small number and difficulty of dataset, lack of problem setups, lack of comparison with recent methods, lack of investigation of parameter sensitivity and computational costs.

Based on these concerns, the authors have provided a very informative rebuttal and follow-up discussions with new results with more datasets (CLCIFAR10, CLCIFAR20, Tiny ImageNet), more setups (biased complementary labels), more SOTA methods (from 2023,2024), parameter sensitivity investigation (k, lambda), computational cost investigation, more insights and discussions regarding the tightness of bounds and key assumptions/conditions used in the paper, and clarifications about the difference between some similar papers or partial-label learning setup.

Perhaps one remaining unaddressed point is experiments with text datasets due to the difficulty of using data augmentation for text. This may be acceptable if a paper can still contribute to areas where data augmentation is easier.

The reviewer scores are 6,5,5,5,5, where all reviewers feel the paper is borderline. Based on the above, we feel many of the initial concerns were addressed, and we recommend acceptance.

**Additional Comments On Reviewer Discussion:**

There were some discussions among the reviewers and the authors after the rebuttal was sent:

Reviewer ocgG: asked a followup question, maintained score 6.

Reviewer K6ZB: did not provide a reply to the rebuttal.

Reviewer Gske: acknowledged that the reviewer read the rebuttal.

Reviewer K5tE: acknowledged that the reviewer read the rebuttal.

Reviewer zFpU: asked followup question, raised score to 5.

At least at the time of writing the meta review, there seemed to be no strong remaining concerns expressed from reviewers, in the reviewer-AC discussions.

---

### Decision · Program_Chairs · 2025-01-22

Accept (Poster)